# Asymmetry and pathways of inter-hemispheric transport in the upper troposphere and lower stratosphere

Xiaolu Yan[1,a], Paul Konopka[2], Marius Hauck[4], Aurélien Podglajen[5], and Felix Ploeger[2,3]

[1]State Key Laboratory of Severe Weather & CMA Key Laboratory of Atmospheric Chemistry, Chinese Academy of Meteorological Sciences, Beijing, China
[2]Institute for Energy and Climate Research: Stratosphere (IEK-7), Forschungszentrum Jülich, Jülich, Germany
[3]Institute for Atmospheric and Environmental Research, University of Wuppertal, Wuppertal, Germany
[4]Institute for Atmospheric and Environmental Sciences, Goethe University Frankfurt am Main, Frankfurt am Main, Germany
[5]Laboratoire de Météorologie Dynamique (LMD/IPSL), École polytechnique, Institut polytechnique de Paris, Sorbonne Université, École normale supérieure, PSL Research University, CNRS, Paris, France
[a]formerly at: Institute for Energy and Climate Research: Stratosphere (IEK-7), Forschungszentrum Jülich, Jülich, Germany

**Correspondence:** Xiaolu Yan (xiaoluyan@cma.gov.cn)

**Abstract.** Inter-hemispheric transport may strongly affect the trace gas composition of the atmosphere, especially in relation to anthropogenic emissions which originate mainly in the Northern Hemisphere. This study investigates the transport from the boundary surface layer of the Northern Hemispheric (NH) extratropics (30-90° N), Southern Hemispheric (SH) extratropics (30-90° S), and tropics (30° S-30° N) into the global upper troposphere and lower stratosphere (UTLS) using simulations with the Chemical Lagrangian Model of the Stratosphere (CLaMS). In particular, we diagnose inter-hemispheric transport in terms of the air mass fractions (AMF), age spectra, and the mean age of air (AoA) calculated for these three source regions. We find that the AMFs from the NH extratropics to the UTLS are about five times larger than the corresponding contributions from the SH extratropics and almost twenty times smaller than those from the tropics. The amplitude of the AMF seasonal variability originating from the NH extratropics is comparable to that from the tropics. The NH and SH extratropics age spectra show much stronger seasonality compared to the seasonality of the tropical age spectra. The transit time of NH extratropical origin air to the SH extratropics is longer than vice versa. The asymmetry of the inter-hemispheric transport is mainly driven by the Asian summer monsoon (ASM). We confirm the important role of ASM and westerly ducts in the inter-hemispheric transport from the NH extratropics to the SH. Furthermore, we find that it is an interplay between the ASM and westerly ducts which triggers such cross-equator transport from boreal summer to fall in the UTLS between 350 K and 370 K.

## 1 Introduction

The transport from the troposphere to the stratosphere plays an important role in determining the chemical composition of the atmosphere, and hence radiative features, which can impact atmospheric chemistry and global climate. For example, ozone-depleting substances (ODS), greenhouse gases, and aerosols in the atmosphere are mainly driven by natural and anthropogenic emissions at the Earth's surface (e.g. WMO, 2018). Although tropospheric air enters the stratosphere mainly through the tropical tropopause layer (TTL) (e.g. Holton et al., 1995; Levine et al., 2007; Fueglistaler et al., 2009), which is then transported to

the global stratosphere by the Brewer-Dobson (BD) circulation (e.g. Butchart, 2014), significant contributions of air mass transport from troposphere to stratosphere are found through other pathways, e.g. monsoons (e.g. Randel et al., 2012; Ploeger et al., 2017), quasi-isentropic transport through the extratropics (e.g. Hoor et al., 2005; Bönisch et al., 2009; Konopka and Pan, 2012), and inter-hemispheric transport (e.g. Tomas and Webster, 1994; Waugh and Polvani, 2000; Orbe et al., 2018; Wu et al., 2018).

Inter-hemispheric transport is important for understanding the distributions of atmospheric tracers because of the asymmetry in anthropogenic emissions between the Southern Hemisphere (SH) and the Northern Hemisphere (NH), with generally strongest emissions in the NH caused by the higher population density. For instance, the anthropogenic and long-lived greenhouse gas $SF_6$ in the atmosphere, which is widely used in the study of transport (e.g. Maiss et al., 1996; Denning et al., 1999; Gloor et al., 2007; Patra et al., 2011; Krol et al., 2018), mostly originates from the NH (e.g. Ravishankara et al., 1993;
Kovács et al., 2017). The ALE/GAGE experiment showed that nearly 95% of the reported sources of ODSs (e.g. $CH_3CCl_3$ and $CFCl_3$) are in the NH with maxima centred around the mid-latitudes (Wang and Shallcross, 2000). Besides anthropogenic emissions, natural emissions in the NH are different from those in the SH due to the asymmetry of topography and land-sea distribution between the two hemispheres. Although the source distributions of many tracers are different in the SH and NH, the observed trends of the tracers are almost homogeneous in the global upper troposphere and lower stratosphere (UTLS), which
suggests the key role of inter-hemispheric transport in regulating the distribution of atmospheric trace gases and maintaining the mass balance (e.g. Müller and Brasseur, 1995; Wang and Shallcross, 2000; Liang et al., 2014; Patra et al., 2014; Chen et al., 2017). Holzer (2009) found that approximately one-third of the NH extratropical surface air is transported to the SH extratropical surface Francey and Frederiksen (2016) emphasised the importance of inter-hemispheric transport inter-annual variations in explaining the sudden increase of the annual mean $CO_2$ difference between Mauna Loa in the NH and Cape Grim in the SH
during 2009-2010. The redistribution of the tracers can strongly affect the dynamical and chemical processes in the atmosphere.

A simplified model including two well mixed boxes respectively for the SH and the NH has been extensively used to quantify the inter-hemispheric transport in previous studies (e.g. Czeplak and Junge, 1975; Denning et al., 1999; Lintner et al., 2004; Patra et al., 2009; Chen et al., 2017; Krol et al., 2018; Naus et al., 2019). Due to the strong transport barrier between the tropics and extratropics (e.g. Hoskins et al., 1985; Bowman, 2006; Kunz et al., 2011), a three-box model including the SH extratropics,
tropics, and NH extratropics was suggested to be used in quantifying inter-hemispheric transport (e.g. Bowman and Carrie, 2002; Erukhimova and Bowman, 2006). The exchange time ($\tau_{ex}$) across the equator is one common parameter to quantify the inter-hemispheric transport, and is defined by the mass balance equation derived from the difference of mean mixing ratios of tracers in the NH and SH and the net cross-equatorial flux (See e.g. Jacob et al., 1987; Patra et al., 2009; Waugh et al., 2013). $\tau_{ex}$ is related to the calculation method and to the hemispheric distribution and emissions of the tracer chosen. Values
of $\tau_{ex}$ range between 0.8-2.0 yr based on the calculations of different models and passive tracers (e.g. Maiss et al., 1996; Denning et al., 1999; Peters et al., 2004; Patra et al., 2011; Liang et al., 2014; Yang et al., 2019).

Age of air (AoA) is another widely used variable to quantify the inter-hemispheric transport, which provides more information compared to the inter-hemispheric exchange time $\tau_{ex}$. AoA can be derived from the age spectrum, and has also been determined from observations of species with a nearly linear growth in mixing ratios such as for $CO_2$ and $SF_6$ (e.g. Hall and Plumb,
1994; Volk et al., 1997; Engel et al., 2009; Stiller et al., 2012; Ray et al., 2014; Engel et al., 2017) and from model simulations

(e.g. Schoeberl et al., 2005; Garny et al., 2014; Konopka et al., 2015; Ploeger and Birner, 2016). These AoA calculations are sensitive to the tracers chosen, to the method of calculation and to the models and reanalysis data used (e.g. Krol et al., 2018; Fritsch et al., 2019; Hauck et al., 2019; Podglajen and Ploeger, 2019; Ploeger et al., 2019). Waugh et al. (2013) estimated AoA using $SF_6$ observations and showed that the mean AoA from the NH midlatitude surface to the SH midlatitudes surface is around 1.4 years. Orbe et al. (2016) and Chen et al. (2017) highlighted the important role of monsoon circulation in reducing mean AoA in the SH with respect to the NH. Konopka et al. (2017) and Krol et al. (2018) recently discussed an interesting asymmetry feature in inter-hemispheric transport with more effective transport from the NH surface to the SH than vice versa.

Many mechanisms have been proposed to drive the inter-hemispheric transport including transport related to eddies and wave breaking (e.g. Czeplak and Junge, 1975; Tomas and Webster, 1994; Staudt et al., 2001), vertical convective transport (e.g. Prather et al., 1987; Hartley and Black, 1995; Denning et al., 1999; Lintner et al., 2004; Erukhimova and Bowman, 2006), seasonal modulation of the Hadley circulation related to the migration of the intertropical convergence zone (ITCZ) across the equator (e.g. Bowman and Cohen, 1997; Wang and Shallcross, 2000), and monsoon circulation (e.g. Orbe et al., 2016; Chen et al., 2017). Zonally resolved results showed that westerly ducts over the Pacific and Atlantic regions are favored regions for the inter-hemispheric transport and redistribution of the atmospheric compositions (e.g. Webster and Holton, 1982; Tomas and Webster, 1994; Waugh and Polvani, 2000; Staudt et al., 2001; Ratnam et al., 2015; Francey and Frederiksen, 2016). The westerly ducts are strongest in boreal winter and affected by the El Niño Southern Oscillation (ENSO), being stronger during La Niña periods and weaker during El Niño periods (e.g. Waugh and Polvani, 2000; Staudt et al., 2001; Dlugokencky et al., 2009; Francey and Frederiksen, 2016; Pandey et al., 2017). Although multiple mechanisms were suggested to be the generators of the inter-hemispheric transport, it is still not clear that the inter-hemispheric transport is driven by each mechanism independently or by the combination of different mechanisms.

Most previous studies have focused on the inter-hemispheric transport from the NH to the SH in the troposphere using a two-box model based on zonally mean results, with relatively few analyses investigating the cross-equatorial transport in the stratosphere (e.g. Lintner, 2003; Patra et al., 2009; Holzer, 2009; Patra et al., 2011). As a complementary to previous studies, we intend to further examine the inter-hemispheric transport in the UTLS because of the global chemical, radiative, and climate effects of atmospheric species in the UTLS. The contributions, pathways, and the mechanism from NH extratropics and SH extratropics to the upper troposphere and stratosphere driven by the inter-hemispheric transport have not been well understood hitherto. Since anthropogenic emissions are mainly produced in the NH, the understanding of transport from the NH to the tropics, the SH, and the global upper troposphere and stratosphere is particularly important. In this study, we address the following questions:

(1) How large are the contributions from the NH extratropics, SH extratropics, and tropics to the upper troposphere and stratosphere, and what is the respective transit time?

(2) Which regions are favored regions (pathways) for the inter-hemispheric transport from the NH extratropics to the SH extratropics?

(3) What is the underlying mechanism in terms of dynamics and circulation, and in relation to the mechanisms proposed in past studies?

We investigate the transport from the NH to the SH and vice versa using the simulations from the three dimensional Chemical Lagrangian Model of the Stratosphere (CLaMS) with the atmospheric source regions divided into three domains (the SH extratropics, the tropics, and the NH extratropics). We quantify the contributions and age spectra from these different regions to the upper troposphere and stratosphere using zonally averaged results for the global view. Particularly, we focus on quantifying the inter-hemispheric transport and characterising the pathways of inter-hemispheric transport by analyzing zonal mean and zonally resolved model output. Section. 2 presents data and methods for our analyses. In Sect. 3, we diagnose the seasonality of transport from different source regions. We explore the pathways of inter-hemispheric transport in Sect. 4 and discuss our findings in Sect. 5 before closing with a summary of the key results in Sect. 6.

## 2 Data and methods

In this study, the surface is divided into 3 boxes to investigate the inter-hemispheric transport, which are the NH extratropics (30-90° N), SH extratropics (30-90° S), and tropics (30° S-30° N), respectively. The threshold of 30.0° is a common choice to separate tropics from extratropics (e.g. Fueglistaler et al., 2011) as it divides the Earth's surface into equal areas of both regions, and further coincides approximately with the horizontal transport barriers of the subtropical jet cores. We calculate age spectra and air mass fraction (AMF) to study transport from the surface of the NH extratropics, SH extratropics, and tropics using the CLaMS model. CLaMS is a Lagrangian chemistry transport model (CTM) with trace gas transport driven by horizontal winds and total diabatic heating rates from reanalysis data (e.g. McKenna et al., 2002; Konopka et al., 2004; Pommrich et al., 2014).

We apply the boundary impulse (time-) evolving response (BIER) approach to calculate the age spectrum $G$ following Ploeger and Birner (2016), which is based on the boundary impulse response (BIR) method (e.g. Holzer et al., 2003; Haine et al., 2008; Li et al., 2012; Orbe et al., 2016), but evolves with time in a transient simulation using quasi-observational winds. Multiple tracer pulses are released in the boundary source region $\Omega_i$, with $i$ labeling the source domain (e.g., NH extratropics, SH extratropics, tropics). The passive tracer with mixing ratio $\chi_i$ at location $r$ and time $t$ related to the mixing ratio $\chi_0(t)$ from the boundary surface of different source regions, which defines the AMF from source regions, can be expressed as (e.g. Waugh and Hall, 2002; Orbe et al., 2013; Ploeger et al., 2019):

$$\chi_i(r,t) = \int_0^\infty d\tau \chi_0(\Omega_i, t-\tau) G(r,t|\Omega_i, t-\tau) \tag{1}$$

The age spectrum is calculated from 120 pulses of inert trace gas species from three source regions, with 40 different species pulsed in each region. These pulse tracers approximate a delta distribution lower boundary condition $\chi_0^j(\Omega_i, t) = \delta(t-t_j)$ with $j$=1,...,40, defining tracer pulses at source times $t_j$. The pulse tracer mixing ratios are set to one in the boundary layer of the source region for 30 days, and are set to zero in the boundary layer outside of the initialization region in every time step. The first 24 different species ($j$=1,...,24) with transit time less than 2 years are pulsed every month. The other 16 different species ($j$=25,...,40) are pulsed every sixth month (e.g., 25th species in the 30th month, 26th species in the 36th month, etc.). Hence, all species have been pulsed after 10 years of model simulations, and are reset to zero in the whole atmosphere and pulsed again subsequently thereafter.

Therefore, the model simulations provide a monthly resolution age spectrum for transit times shorter than two years and a semi-annual resolution age spectrum for longer transit times. The integration of the spectrum over time generally yields a value less than 1 and AoA is young-biased caused by the truncation of the simulations at 10 years. Therefore, we calculate the mean AoA for each source region by normalizing the age spectrum to unit norm:

$$\Gamma_i(r,t) = \int_0^{10} \tau G(r,t|\Omega_i,\tau)d\tau / \int_0^{10} G(r,t|\Omega_i,\tau)d\tau \tag{2}$$

The details about the model setup and the calculation of age spectra from multiple pulse tracers and the mean AoA from the age spectrum can be found in Ploeger and Birner (2016) and Ploeger et al. (2019). For the purpose of our study, the pulse tracer mixing ratios are set to zero in the boundary layer during every time step, which allows us to separate the transport contributions and pathways in the UTLS from those of transport in the boundary layer. As inter-hemispheric transport in the lower troposphere has been the focus of several past studies (e.g. Staudt et al., 2001; Orbe et al., 2015; Chen et al., 2017; Krol et al., 2018), here we focus on transport in the UTLS. Detailed comparisons with previous studies using different model setups, which are mainly focused on the inter-hemispheric transport in the troposphere (e.g. Orbe et al., 2015, 2016), will be provided later. For this study, we carried out a simulation covering the period from 1989 to 2017 with transport driven by the meteorological data from ERA-Interim (Dee et al., 2011). Due to the 10 year spin-up time for the age spectra, the model data from 1999-2017 is analyzed in the following to address the questions raised in the introduction.

## 3 Seasonality of transport

### 3.1 Seasonality of air mass fractions

To evaluate the global contributions from the source regions, we calculate the zonally averaged seasonal mean AMFs from the boundary layer of the three source regions. In the following, we use the abbreviations of months (DJF, MAM, JJA, and SON) to represent different seasons. Figure 1 shows the seasonal variations in AMF originating from the NH extratropics, SH extratropics, and tropics during 1999-2017. The sum of the AMF over all 3 source regions is ∼1 related to the limitation of the maximal transit time in our simulations. Note that different colorbars are used for the AMF transported from each source region. The global results show that the AMFs from the NH extratropics to the troposphere and stratosphere are about five times larger than the corresponding contributions from the SH extratropics. The relative contributions of transport from the NH extratropics to the atmosphere compared to those from the tropics depend on the altitude, and they are about ten to fifteen times smaller than the tropical AMFs in the middle and upper troposphere and around 20-40 times smaller in the stratosphere. Although the contributions from the tropics to the UTLS are much larger than those from the NH extratropics, the annual amplitude of tropical AMFs in the UTLS is comparable to that of NH extratropical AMFs related to the small contributions from the SH extratropics.

Newly pulsed air masses (younger than 3 months) from the NH extratropics start to cross the subtropical tropopause in boreal summer (JJA, Fig. 1g). Three months later, air masses from the NH extratropics are elevated to the lower stratosphere first

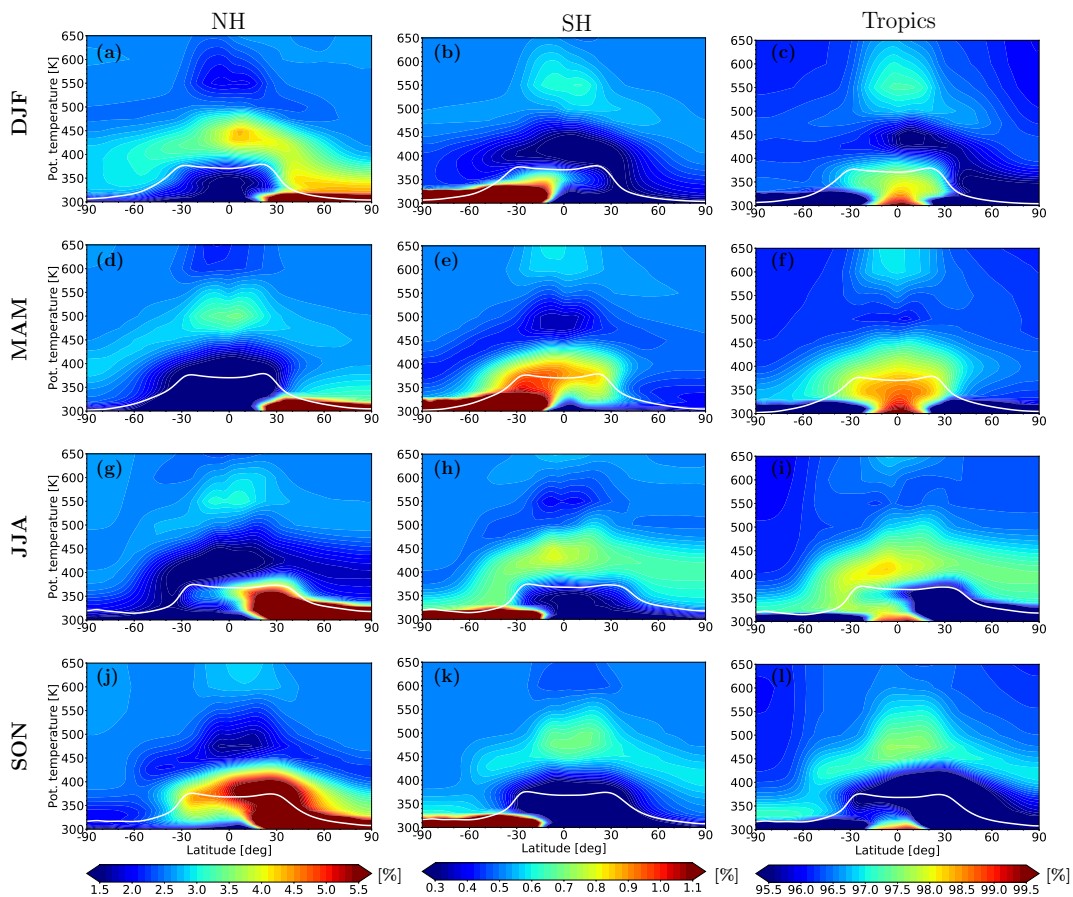

**Figure 1.** Climatological (1999-2017) zonal mean AMFs originated from the NH extratropics (left), SH extratropics (middle), and tropics (right) for different seasons (rows). The AMFs for all 3 source regions add up to ~1. The white line shows the WMO tropopause.

mainly in the Asian summer monsoon (ASM) region driven by the monsoon circulation, and are then transported isentropically
to the tropical lower stratosphere and NH extratropical lower stratosphere covering the latitude range from 30° S up to the Arctic regions (Fig. 1j). Later on, the NH extratropical air masses in the upper tropospheric and lower stratospheric tropics driven by the ASM are further transported to the tropical pipe and the whole SH in DJF and MAM (Fig. 1a and Fig. 1d). Note that young air masses pulsed during boreal winter and spring (DJF and MAM) are not transported to the subtropical stratosphere.

The seasonality in the transport patterns of AMFs originating from the SH extratropics are shifted by 6 months compared to
those from the NH extratropics. Although the respective contributions of the SH extratropics (i.e. in DJF and MAM) show some similarities to those from the NH extratropics (i.e. in JJA and SON), there are few significant differences between transport from NH extratropics and SH extratropics. Crossing of the subtropical tropopause for SH extratropical origin air happens first in austral autumn (MAM, Fig. 1e) rather than austral summer (DJF, Fig. 1b), and the overall impact of the SH extratropical

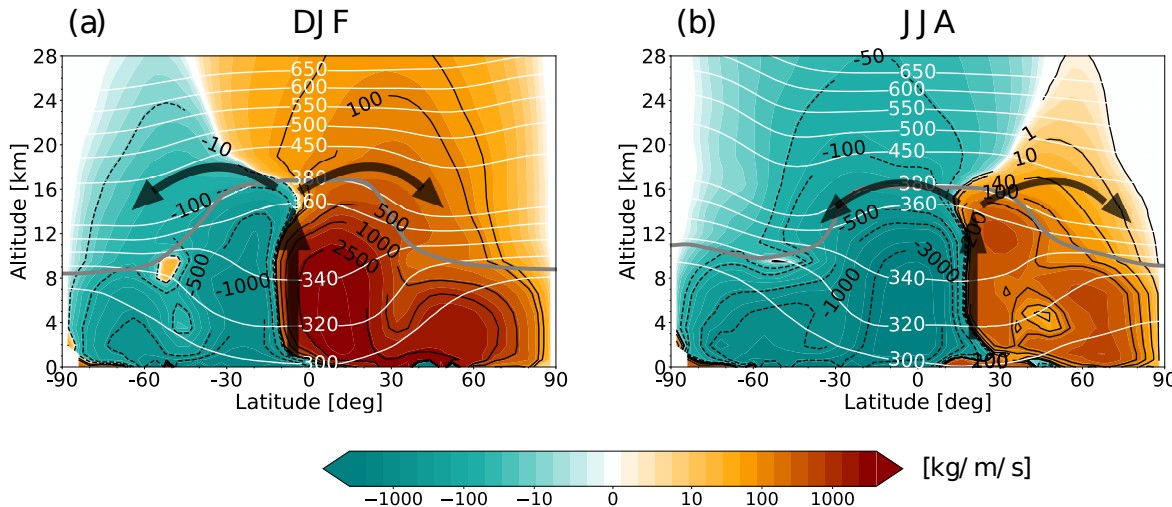

**Figure 2.** Climatology (1999-2017) of the residual mean mass streamfunction (colour shading with a subset of values highlighted in black contours). The grey line shows the WMO tropopause. White contours show potential temperature levels in Kelvins. The thick black arrows illustrate the upwelling of the Hadley circulation and the shallow branch of the BD circulation (around 400 K).

boundary surface tracers on both the tropics and the high latitudes is significantly weaker. Most transport of SH extratropical
origin tracers is inhibited by the tropopause in the subtropical SH during DJF (Fig. 1b) and MAM (Fig. 1e). Especially, the
SH extratropical AMF in the SH lower stratosphere during austral autumn (MAM) is much smaller than the NH extratropical
AMF in the NH lower stratosphere during boreal autumn (SON). These differences are most likely attributed to hemispheric
differences in the strengths of the monsoons (e.g. Orbe et al., 2016; Chen et al., 2017) and in the strength and downward extent
of the polar vortices.
Figure 1(right) shows that the tropical surface air dominates the atmospheric composition in the global UTLS. The season-
ality of the tropical contribution results from the superposition of the Hadley and BD circulations, which are schematically
illustrated in Fig. 2 for DJF (a) and JJA (b) by using the residual mean mass streamfunction. The upwelling of the Hadley and
BD circulations is shifted northward from DJF to JJA. Note the hemispheric asymmetric upwelling positions of the circula-
tions in winter and summer. The Hadley cell upwelling is located in the SH tropics around 5° S during boreal winter, while the
175 upwelling is shifted far into the NH subtropics to latitudes of around 20° N in boreal summer. Thus, the strongest contribution
of the tropical surface air to the stratosphere starts in boreal winter (DJF, Fig. 1c) and peaks in MAM (Fig. 1f) as the result of
strongest positive coupling between the Hadley and BD circulations. The seasonality, in terms of relative amplitude, of the air
originating from the tropics is less pronounced compared to that from the NH extratropics and SH extratropics. The evolution
of the tropical source air shows similar patterns to those from the SH extratropics. The transport pattern is significantly different
from the transport of the NH extratropical source air. It is especially remarkable how the northward shift in the boreal summer
Hadley cell weakens upward transport from the tropics during JJA and SON (Fig. 1i and Fig. 1l) while favouring upward

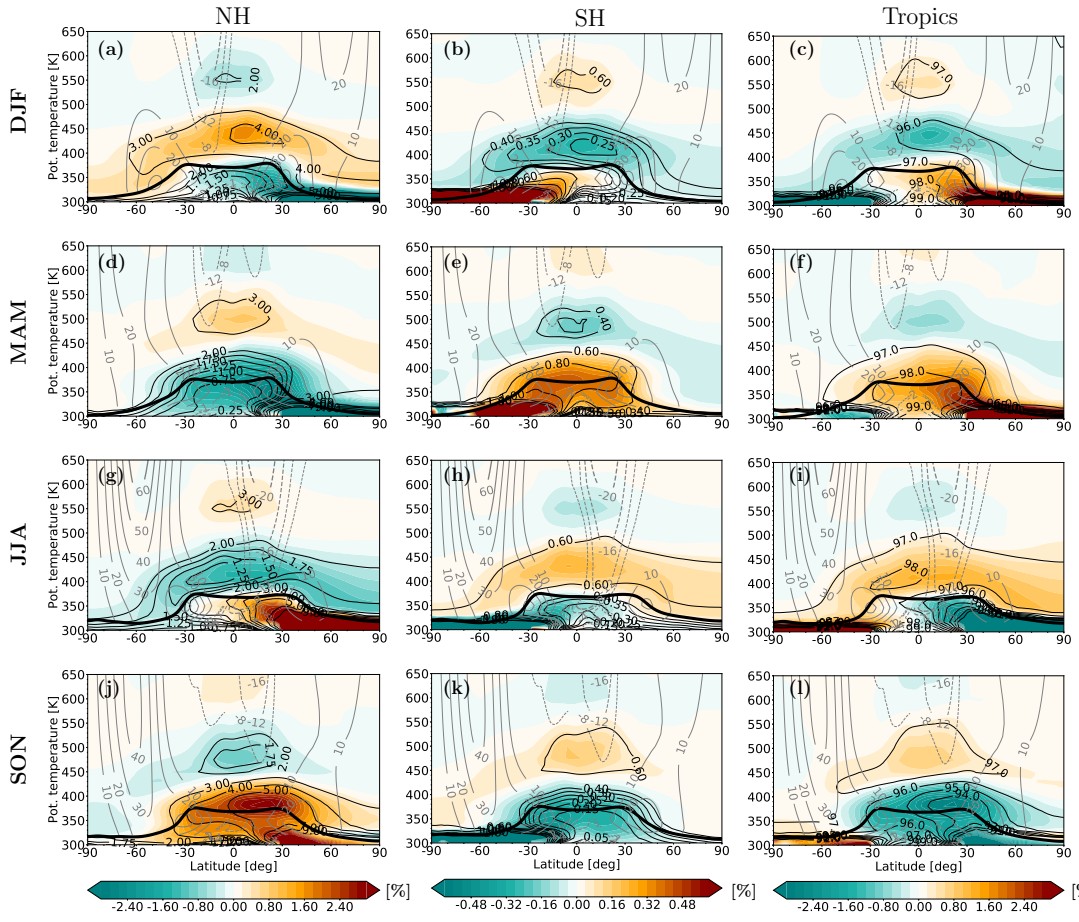

**Figure 3.** Climatological (1999-2017) zonal mean AMF departure from the annual average. The absolute AMF contributions from Fig. 1 are shown as black contours. Grey contours show the mean zonal winds. The thick black line is the WMO tropopause.

transport from the NH surface (Fig. 2b), whereas this effect is much weaker during the austral summer (DJF and MAM, Fig. 1c and Fig. 1f).

To further explore the seasonal variations of transport from the boundary layer, we remove the annual mean of the contribu-
185 tions from each source region. These seasonal anomalies of climatological zonal mean AMF are shown in Fig. 3 respectively from the NH extratropics (left), SH extratropics (middle), and tropics (right). Note the different colorbar for NH extratropical air and SH extratropical air. Clearly, the amplitude of the AMF seasonal cycle in the global UTLS originating from the NH extratropics is comparable to that from the tropics, and about five times larger than that from the SH extratropics. The transport features of the NH extratropical AMF anomalies starting from boreal summer (JJA) again show similar patterns to those from
190 the SH extratropics starting from austral summer (DJF) with a shift of 6 months and much smaller absolute anomalies although the relative anomalies are comparable.

There are also few new structures in Fig. 3 compared to the absolute contributions in Fig. 1. A pronounced positive anomaly in the lower stratosphere over the NH extratropics and tropics in SON (Fig. 3j) is related to the isentropic transport directly above the ASM region after the elevation of NH extratropical air by the monsoon circulation, which again suggests that the tropospheric air in the NH extratropics is mainly transported to the NH and tropical lower stratosphere via the ASM circulation. The positive anomaly of NH extratropical air also crosses the equator and extends southward into the SH subtropics from the middle troposphere to the stratosphere. We will further explore the mechanism to drive the inter-hemispheric transport in the UTLS using zonally resolved data later. Another striking feature is the negative anomaly of the NH extratropical air mass in the layer around 320 K in the NH during boreal autumn (Fig. 3j). This signature might be associated with the combination of less convective activity in boreal autumn in the NH extratropics, or with the suppression of horizontal transport from the subtropical troposphere in the layer around 320 K or, finally, with the southward movement of the Hadley cell enhancing isentropic, poleward transport from the tropics and across the still weak summer-autumn jet (see Fig. 3l).

In contrast to the NH case, the anomaly for SH extratropical air masses shows negative values almost throughout the SH extratropical lower stratosphere during austral autumn (Fig. 3e). This difference to the NH extratropical air (Fig. 3j) is mainly related to the weak convection and the strong inhibition (strong zonal jet) of horizontal transport from the subtropical region in the SH during austral autumn (MAM). In the NH, the tropopause barrier is weak and upward motion over the ASM region is strong, and a substantial amount of NH extratropical origin air can be transported to the lower stratosphere driven by monsoon circulation.

## 3.2 Seasonality of age spectrum and age of air

In Sec. 3.1, we have quantified transport using the AMF, which measures the contribution from different source regions to the air composition in the UTLS. In this section, we provide a complementary view of transport in terms of the age spectrum derived from the same simulations and for the same source regions as in Sec. 3.1. Figures 1 and 3 show that strong isentropic transport across the tropopause occurs in the layer around 360 K. Hence, we consider the age spectrum at 360 K as a reference location for the UTLS.

The age spectra of air from the NH extratropics, SH extratropics, and tropics are illustrated in Fig. 4. Note the different colorbars used for the three source regions. The transport seasonality is evident for the NH extratropics and SH extratropics origin air, even stronger than the seasonality for the age spectra with tropical origin, which is consistent with the results based on the AMF (Fig. 1). The first peak of the NH extratropical age spectrum during boreal summer and autumn (Fig. 4g and Fig. 4j) is a very strong signature compared to the SH extratropical age spectrum during austral summer and autumn (Fig. 4b and Fig. 4e), which means that much more young air can be expected in the NH compared to the SH. The age spectrum of NH extratropical origin air always shows large PDF values at young transit times during boreal summer, and nearly zero during boreal winter, which suggests that the pollutants from the NH extratropics are transported to the global UTLS primarily during boreal summer.

Age spectrum and mean AoA from the SH extratropics show a lot of similarities to those from the NH extratropics shifted by 6 months. However, NH extratropical origin young air (< 6 months) shows peak values around the ASM region (Fig. 4g and

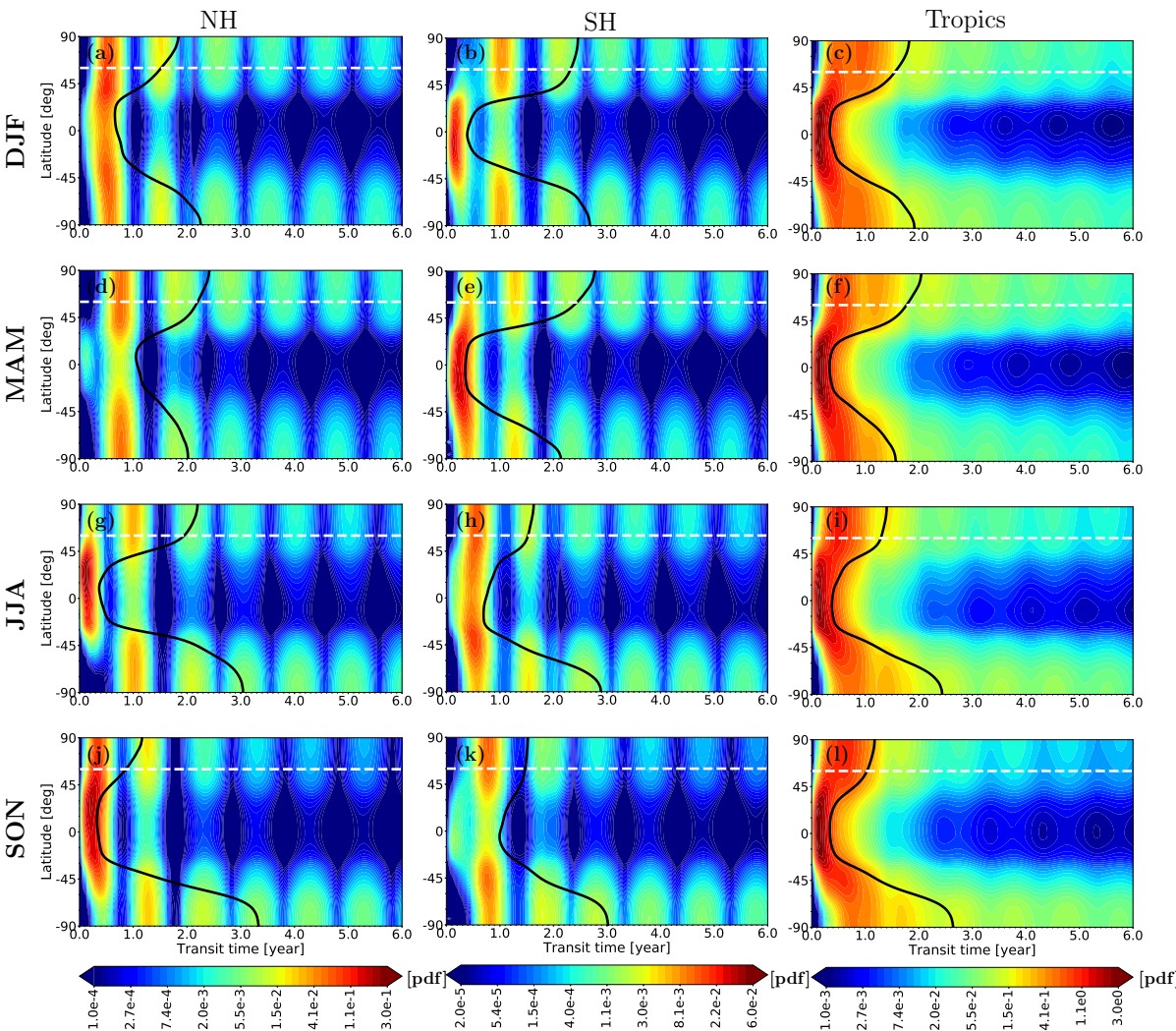

**Figure 4.** Age spectra as (partial) transit time probability density functions (PDFs), calculated for air originating from three source regions: NH extratropics, SH extratropics, and tropics. Age spectra are shown at $\theta = 360\,\mathrm{K}$ (destination region) for all seasons. For each season, the sum of the integral of the age spectra over all 3 source regions (i.e. over all partial contributions) is $\sim$1, with deviations caused by our approximation limiting the maximal transit time to 10 years. The black line shows the mean AoA as derived from the age spectrum. White dashed line marks the 60° N latitude.

Fig. 4j). The flushing of the NH lowermost stratosphere with NH extratropical origin air during boreal summer and autumn (JJA and SON) is more pronounced compared to the flushing of the SH lowermost stratosphere with SH extratropical origin air during austral summer and autumn (DJF and MAM). The mean AoA shows that inter-hemispheric transport proceeds faster from the SH extratropics to the NH extratropics than from the NH extratropics to the SH extratropics in qualitative agreement

with results found by Konopka et al. (2017, their Fig. 5) related to the weaker barrier along the jet in the NH (see Fig. 3) which allows faster horizontal transport. Another important asymmetry is that, with exception of MAM, the mean AoA is always older in the SH than in the NH for all other seasons and for all source tracers. This is mainly a consequence of hemispheric differences in the wave-driven eddy mixing, being stronger in the NH throughout the year (Rosenlof, 1995; Konopka et al., 2015).

Figure 5 confines the global age spectrum shown in Fig. 4 to partial age spectra at the latitude of 60° N, which defines the spectra from individual source region without normalization to 1. The age spectrum for the NH extratropical origin air (Fig. 5a) shows multiple peaks caused primarily by the interplay between Hadley and BD circulations. The Hadley cell upwelling is shifted to the NH subtropics during boreal summer, which is the season favoring upward transport from the NH surface, and peaks in the spectrum are related to air originating at the NH surface in early summer. The youngest peak is in JJA at

transit times of around 2 months as a result of an "in phase" interaction between the Hadley and the lower branch of the BD circulation. The respective first peaks in the following seasons are shifted accordingly, which suggests that most of air in the NH high latitude region with origin in the NH extratropical boundary layer is emitted during boreal early summer. Although the tropical upwelling has its maximum in boreal winter and spring, it does not significantly transport the NH extratropical origin air to the high latitude lower stratosphere. This is mainly because the Hadley cell supports such transport pathway rather

in summer than in winter and spring (Fig. 2). In addition, transport of the NH extratropical origin air to the high latitudes maximizes in boreal autumn. Note that the second peak in JJA resulting from the NH extropical origin air is higher than the first peak. The mean AoA shows youngest value in boreal autumn (SON) and oldest value in boreal spring (MAM).

Although the structure of age spectrum of the SH extratropical origin air (Fig. 5b) also includes multiple peaks like that from NH extratropics, its total contribution is almost 10 times smaller than the respective contribution from the NH extratropics. The

250 first peak in age spectra in each season from MAM to DJF is delayed by around 3 months accordingly, which again suggests that the main contribution from SH extratropics originates in austral summer. The mean AoA from the SH extratropics is older than that from the NH extratropics during each season except in JJA.

The age spectrum of tropical origin (Fig. 5c) shows by far the highest partial contribution (10 and 100 times larger than that of the NH and SH, respectively). Unlike the age spectrum of the NH extratropics and SH extratropics origin air, the tropical

age spectrum in JJA and SON has only one clear peak at transit time around 6 months. During DJF and MAM the tropical age spectrum shows more a multimodal shape with primary peak at transit time around 6 months and a secondary peak delayed by a few months. The first peak might be related to the Hadley and BD circulations combining with the rapid isentropic transport. The second peak shows similar transit time as the air originating from the SH extratropics, which suggests that the peak might be driven by recirculation within the shallow branch of the BD circulation. The age spectra along 60° S on the 360 K isentropic

surface shows similar patterns with 6 months shift and different amplitudes (not shown).

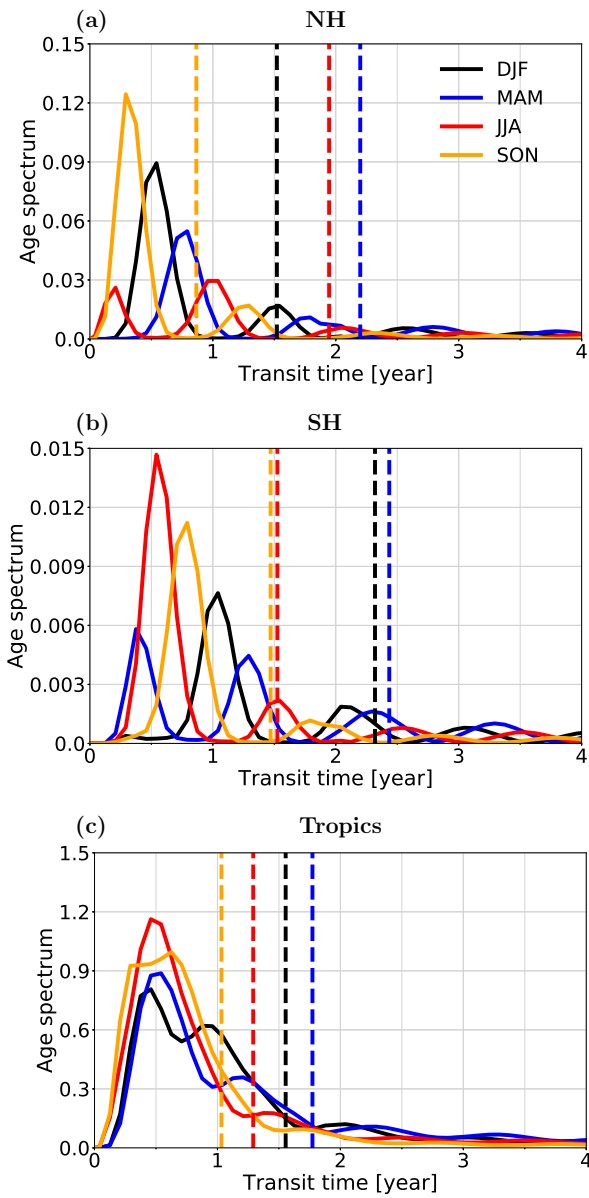

**Figure 5.** Partial age spectra of the air with the origin in: NH extratropics (a), SH extratropics (b), and tropics (c) calculated at 360 K along 60° N (destination region). Different colours represent different seasons. Vertical dashed lines indicate the mean AoA. Note different ranges of the y-axis (Tropics = 10×NH, NH = 10×SH) which quantify relative differences of the considered source regions.)

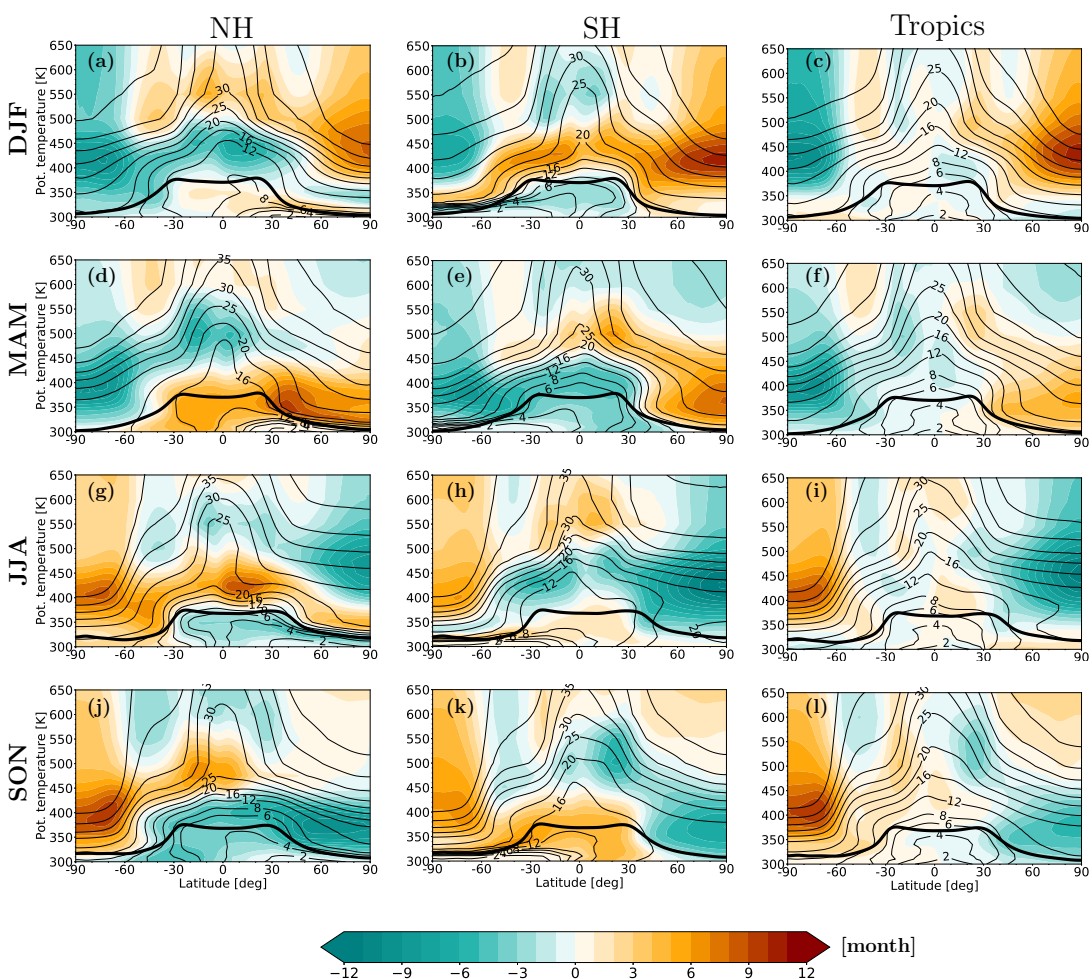

**Figure 6.** Climatology of the mean AoA (1999-2017, black contours) and the AoA anomaly with respect to this climatology (colour shading) from the NH extratropics, SH extratropics, and tropics for each season. The black line shows the WMO tropopause.

Finally, to get the global view of the mean transit time seasonality from the three source regions, we calculate for each season the mean AoA from the respective age spectrum using Eq. 2. The mean AoA for each source region and season with and without the annual mean removed are compared in Figure 6. A similar picture emerges on the well-known seasonality and hemispheric asymmetries of the Hadley and BD circulations (e.g. Konopka et al., 2015, and citations therein). While the seasonality of the tropical upwelling dominates the tropical features, the strength of isentropic poleward transport and polar vortices explain the patterns at high latitudes. The highest amplitude in the AoA anomalies for NH sources can be diagnosed in the polar SH and vice versa for the SH sources in the polar NH. The amplitude of seasonality in the tropics, especially of the air originating in the tropics, is much smaller compared to that in the NH extratropics and SH extratropics.

The tropospheric air originating from the NH extratropics shows younger mean AoA during boreal summer (JJA) mainly in the NH (negative anomalies in Fig. 6g) leading to the flushing with fresh air and the propagation of young air upward and southward to the global stratosphere in the following seasons. Mean AoA patterns from the SH extratropics show a lot of similarities to those from the NH extratropics with a 6 months shift. The young tropospheric air originating from the SH extratropics starts filling the UTLS from austral summer (DJF) in the SH and is transported upwards and northwards in the following seasons. Besides the similarities, SH extratropical origin air shows an old layer (about 20 months) around the NH extratropical tropopause (around the altitude range of 320-350 K) during JJA (Fig. 6h) linked to the flushing of the NH lowermost stratosphere with NH extratropical air (Fig. 1g). This old layer also exists in the distribution of tropical origin air over high latitude regions (Fig. 6i).

Beyond these known features, some interesting asymmetries of the cross-hemispheric transport can be diagnosed. Comparing the left and middle column of Fig. 6, we find that the age of SH extratropical origin air in the NH is younger than the age of NH extratropical origin air in the SH associated with the fast flushing of the NH lower stratosphere with young air in summer (Hegglin and Shepherd, 2007; Bönisch et al., 2009; Orbe et al., 2016; Konopka et al., 2017). The latitudinal mean AoA gradients in the tropical upper troposphere are weak because of the increased latitudinal transport caused by the upper branch of the Hadley circulation and isentropic mixing. The latitudinal mean AoA gradients of NH extratropical tracer in the SH during JJA are larger than those of SH extratropical tracer in the NH during DJF caused by the stronger barrier along the jet in the SH during austral winter.

## 4 Pathways of inter-hemispheric transport

In Sect. 3, we discussed the transport from the three source regions based on zonal mean results. Clear hemispheric asymmetry features of transport were noticed in AMFs and age spectra. The transit time from the SH extratropical surface to the NH is shorter than that from the NH extratropical surface to the SH. The contributions (AMFs) of the NH extratropical air to the global UTLS are around 5 times larger than those from the SH associated with the stronger monsoons and weaker transport barriers in the NH during boreal summer, which allow strong meridional and inter-hemispheric transport. To gain deeper insights into these hemispheric asymmetries in transport, we disentangle the transport pathways in this section using zonally resolved data. Since most of the anthropogenic pollutants are emitted in the NH and the contributions from the NH extratropics to the SH are much larger than vice versa, the transport pathways from the NH to the SH are of our particular interest.

The monthly evolution of young air (AoA less than 3 months) from the NH extratropics along the latitude of $10°$ S longitude-pressure cross-section is illustrated in Fig. 7. We choose $10°$ S to reduce the influence of reversible transport across the equator. The AMF from January to May is nearly zero and therefore only May is shown (Fig. 7a), which is different from the previous results (e.g. Holzer 2009, their Fig. 3 and Orbe et al. 2016, their Fig. 5a-e). This difference is likely caused by the model setup in our simulations with the species set to zero in the boundary layer outside of the source region in each time step, which eliminates the cross-equatorial transport from NH extratropical surface to the ITCZ region in the boundary layer and then ascent over the tropics and SH subtropics. The convection in boreal winter is not strong enough to lift abundant NH

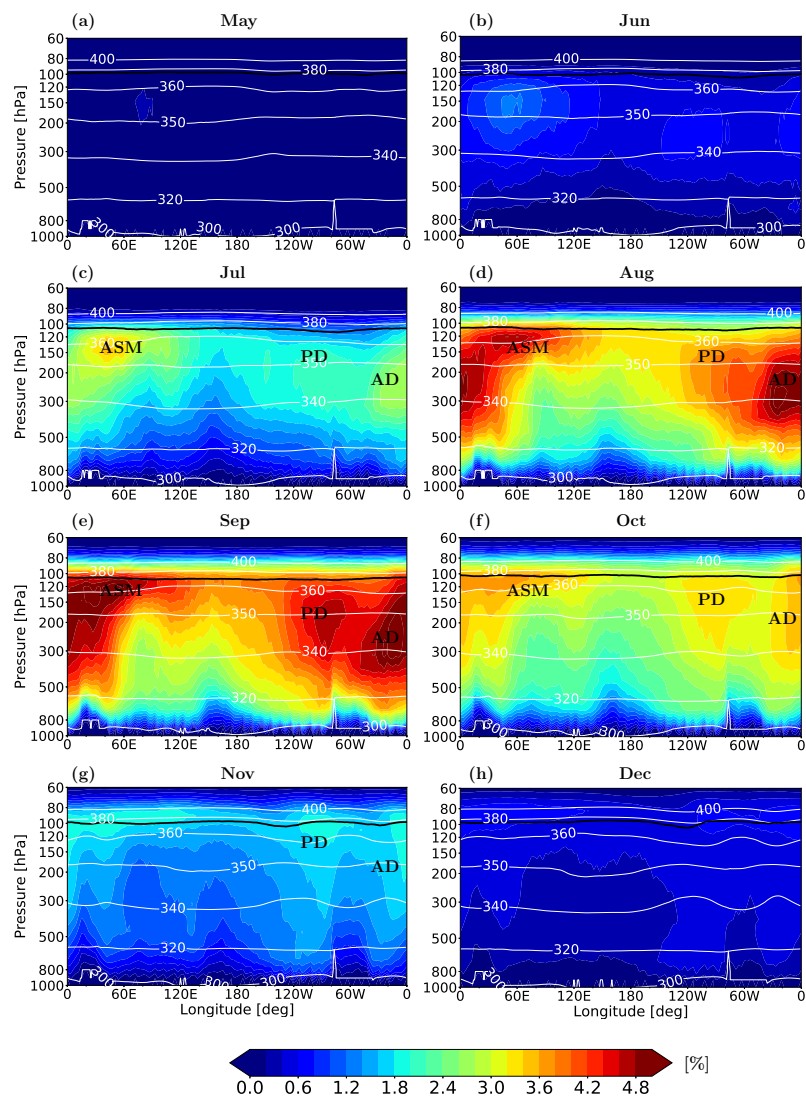

**Figure 7.** Longitude-pressure cross-section along the latitude of 10° S of monthly mean NH extratropical young (< 3 months) AMF (colour shading) during 1999-2017. White lines show isentropic levels, the black line the WMO tropopause. ASM, PD, and AD respectively indicate the rough locations of the Asian summer monsoon region, Pacific westerly ducts, and the Atlantic westerly ducts.

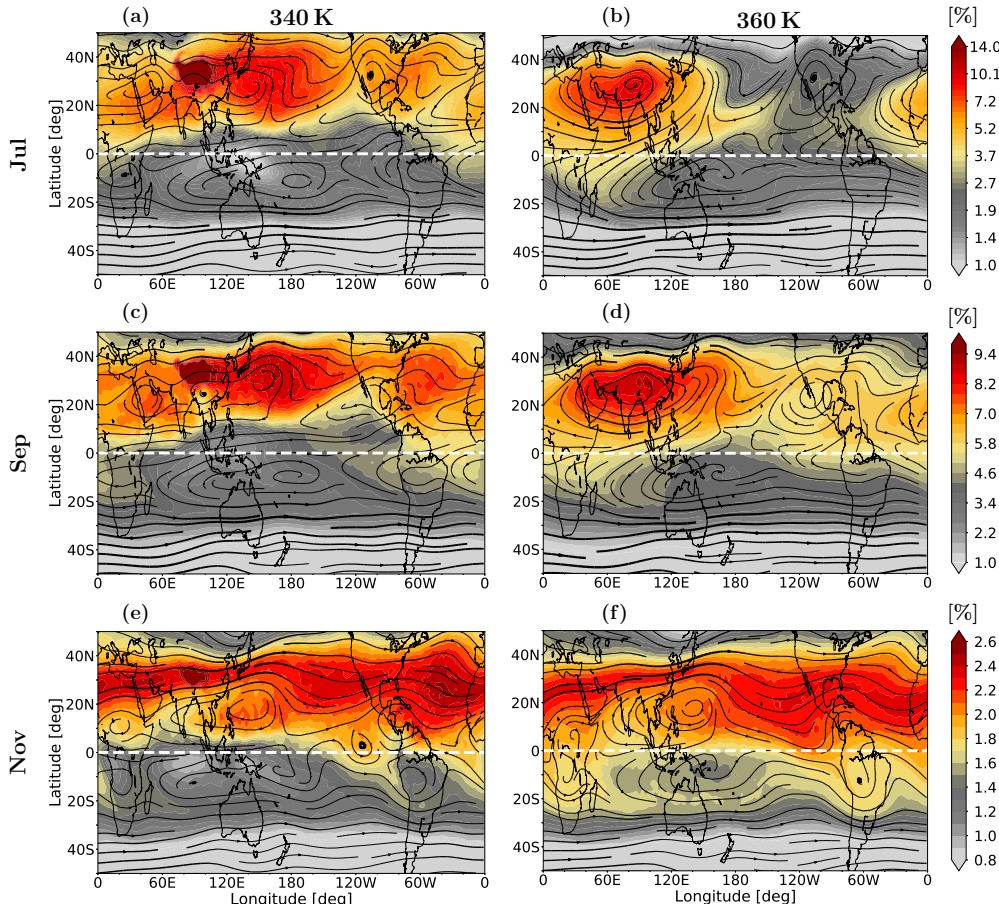

**Figure 8.** A snapshot of the horizontal distribution of the NH extratropical origin young (< 3 months) air on the 340 K and 360 K isentropic surface during July, September, and November. Streamlines show horizontal winds.

extratropical air to high altitude for isentropical transport in the UTLS, which suggests the lack of inter-hemispheric transport driven by convective divergent outflow during winter. First significant signatures become evident in June−July, maximize in August−September, and vanish in November. There is almost no inter-hemispheric exchange in the lower troposphere. Despite the model setup, this is probably related to stable easterlies, which are less disturbed by Kelvin waves and which effectively act as a meridional transport barrier.

Young air masses from the NH extratropics are transported to the SH from June to October (Fig. 7b-f) mainly along two distinct pathways: between 0 and 120° E (denoted as ASM region) in the altitude range of 340-390 K as well as above the Atlantic (around 20° W) and Pacific (around 80° W), denoted as regions of westerly ducts (AD/PD for Atlantic and Pacific ducts, respectively). The cross-equatorial transport over the Atlantic and Pacific peaks at lower altitudes mainly in the upper troposphere between 340 and 350 K compared to the cross-hemispheric transport over ASM region. The relative importance

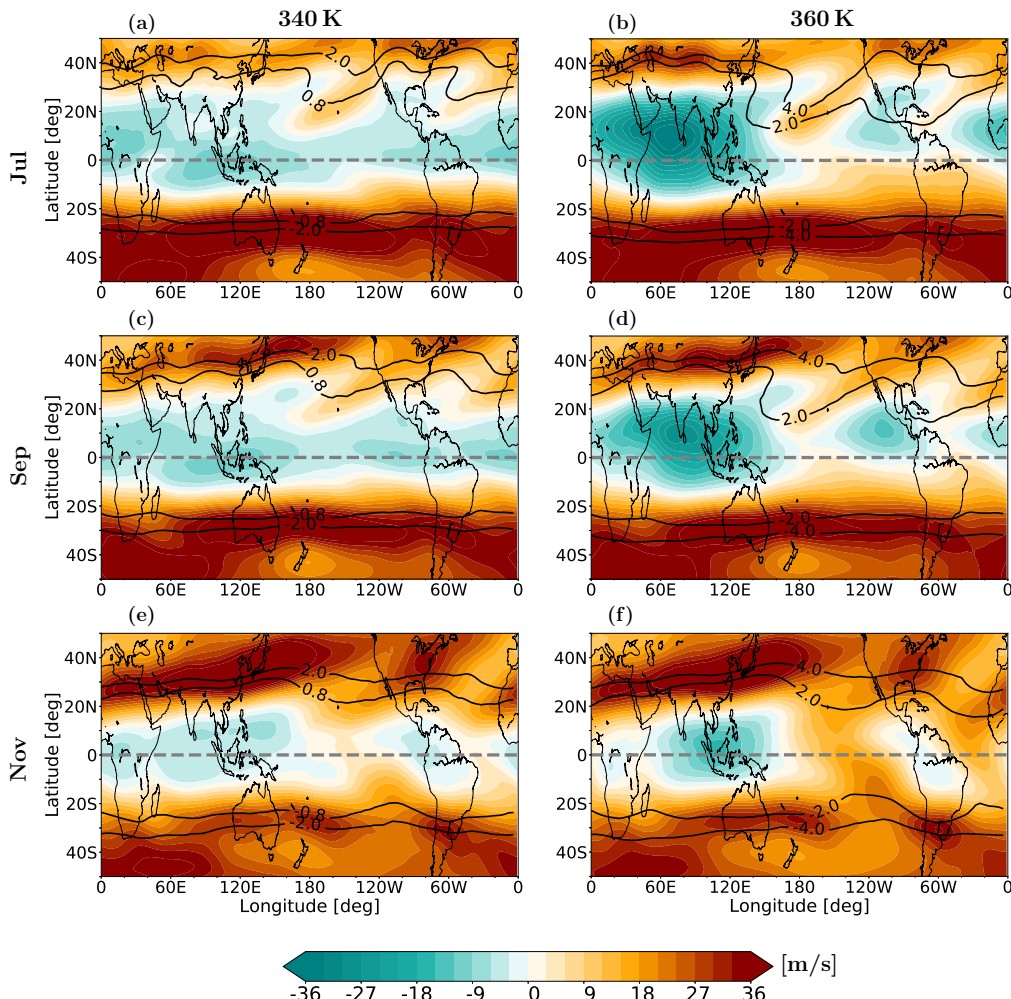

**Figure 9.** Climatological (1999-2017) horizontal distribution of zonal wind on the 340 K and 360 K isentropic surface during July, September, and November. The potential vorticity is indicated by the black contours.

of the inter-hemispheric transport through the ASM pathway compared to those over Atlantic and Pacific seems to be stronger in our study than in Orbe et al. (2016, c.f. their Fig. 5f-j). To disentangle which flow properties in the NH cause this pattern of inter-hemispheric exchange, zonally revolved views of both the AMF (less than 3 months) and the zonal wind overplotted with PV contours are shown in Fig. 8 and Fig. 9, respectively. Here, monthly means (July, September, and November) of young AMF are plotted at 340 and 360 K potential temperature levels. We still stick to our simplified notation (UTLS), even a significant part of transport occurs within the TTL.

Figure 8 shows that the time evolution of the AMF crossing the equator is caused by the combination of two dynamical processes: (I) ASM anticyclonic flow and related eddy shedding, mainly at $\theta = 360$ K (Popovic and Plumb, 2001; Orbe et al., 2016) and (II) the eastward-propagating Rossby wave dynamics across the equator in regions of westerly winds (westerly ducts), mainly over the Pacific and the Atlantic. In July, the peak of the young air (less than 3 months) at 340 K is located over Tibetan plateau and is related to the elevated orography over Tibet which is very close to the 340 K level, so the peak is strongly affected by the released boundary tracers. A second peak is located in the subtropics of Western Pacific and can be attributed to the outflow from monsoon circulation at lower level. The ASM circulation keeps supplying the NH extratropical young air from lower level (Fig. 8a and Fig. 8c) and isolates most of the young air inside the center of the ASM anticyclone at 360 K (Fig. 8b and Fig. 8d) during July-September. Part of the NH extratropical origin air which was entrained into the ASM anticyclone moves southward and westward along with the ASM circulation, and is then transported to the SH by eddy shedding detaching ASM air from the anticyclone and subsequently being transported into the SH (Orbe et al., 2016, their Fig. 7) and to the Atlantic by the easterly flow on the southern edge of the ASM anticyclone.

The westerly ducts can be clearly seen in the respective climatology of the zonal wind and PV shown in Fig. 9. The existence of westerly ducts is altitude dependent. The tongues of PV and the related anomalies of the westerly wind can be diagnosed in the NH, both over the Pacific and the Atlantic, from July to November at the potential temperature level of 360 K, while at 340 K westerly ducts only appear in boreal autumn and winter (starting from October, not shown). Note that the westerlies in the NH become stronger from July to November. The impact of the westerly ducts on the cross-hemispheric transport can be deduced from the time evolution of the AMF at 340 K (Fig. 8 left) with some distinct signatures over the Atlantic and slightly weaker signatures over the Pacific. Although the westerly ducts don't exist in July and September at 340 K, weak westerlies at levels below allow cross-equatorial transport over the Atlantic.

The picture changes at $\theta = 360$ K (Fig. 8 right). While the eddy shedding mechanism plays an important role from July to September, there is only weak transport from the NH to the tropics and to the SH through the westerly ducts during this time. This implies the important role of the ASM circulation in the asymmetry of inter-hemispheric transport at the 360 K level. However, starting from September, the westerly ducts start to drive the cross-hemispheric transport (Fig. 8d and Fig. 8f). On the one hand, the westerly ducts are getting stronger in boreal autumn and winter (Fig. 9d and Fig. 9f) compared to boreal summer (Fig. 9b). On the other hand, most of the tracers transported across the equator through the westerly ducts originate from ASM regions following the evolution of the ASM anticyclone. This suggests that the westerly ducts alone would not transport a substantial amount of young air masses from the NH extratropics to the SH. Significant inter-hemispheric transport through westerly ducts only happens when the NH extratropical air has been transported into the UTLS by the ASM aforehand. Hence, it is the interplay between the ASM anticyclone and the westerly ducts which drives the inter-hemispheric transport from boreal summer to autumn.

The monthly evolution of zonal winds along the equator and young AMF from NH extratropics over the region of [6° N, 20° N] are illustrated in Fig. 10. Only the results from every second month are shown here to reduce the redundancy. Westerly ducts in the UTLS (350-380 K) are strongest in January and weakest in September. However, we find that the largest inter-hemispheric transport happens in September (Fig. 7e) with neither the strongest ASM nor the strongest westerly ducts.

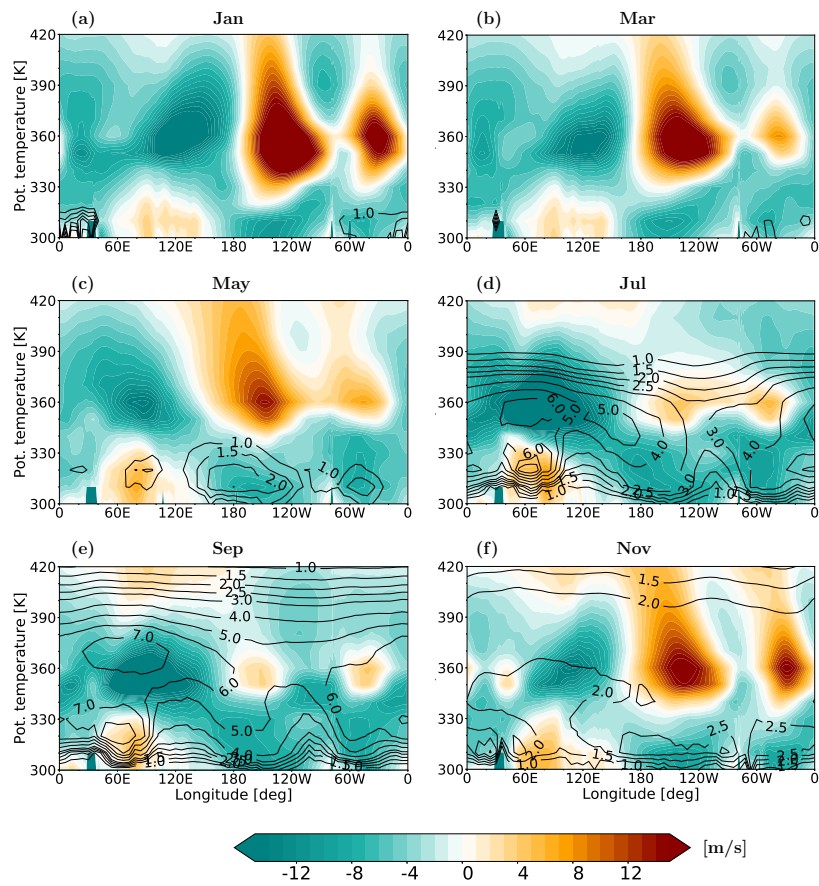

**Figure 10.** Longitude-potential temperature cross-section along the equator of monthly mean zonal wind overplotted with the monthly zonal mean of NH extratropical young (< 3 months) AMF (contours) over the domain of [6° N, 20° N] during 1999-2017.

Figure. 10 shows that the largest amount of NH extratropical air (highest AMFs) is transported to the UTLS over southern ASM region, Pacific, and Atlantic in September by the ASM circulation and eddy shedding before crossing the equator. And they are larger than the AMFs in July (strongest ASM) and in January (strongest westerly ducts), which again suggests that neither the westerly ducts nor the ASM alone determines cross-equatorial transport. We examine the monthly evolution of young (< 3 months) AMFs from NH extratropics on the isentropic surface between 340 K and 420 K from May to December (not shown) and notice that the coupling effect between ASM and westerly is time and altitude dependent. The interaction between ASM and westerly ducts mainly drives the inter-hemispheric transport during autumn in the UTLS due to the coincidence of the westerly ducts and a significant amount of NH air at UTLS levels, which was transported upwards by the ASM during the previous months. This coupling effect plays an important role in the inter-hemispheric transport from summer to autumn at the altitude level between 350 K and 370 K.

## 5   Discussion

The air contributions and age spectrum (or AoA) from different source regions to the destination regions in the atmosphere
provide valuable information for understanding the effect of natural and anthropogenic emissions on the atmospheric compo-
sition and climate. However, recent studies show substantial transport uncertainties depending on the used methods, models,
and meteorological reanalyses (e.g. Krol et al., 2018; Ploeger et al., 2019).

Recently, Hauck et al. (2020) estimated the age spectra and AMF also using simulations from CLaMS but with the pulse
tracers released from the tropopause level over the NH extratropics (30-90° N), SH extratropics (30-90° S), and tropics (30° S-
30° N). The first obvious difference to our results is that the NH extratropics and SH extratropics origin air pulsed in the
boundary layer contributes much less to the lower stratosphere compared to air originating at tropopause level (Hauck et al.,
2020, their Fig. 2). However, the tracers pulsed at the tropopause level are more latitudinally confined compared to the results
here. Furthermore, the inter-hemispheric transport is more symmetric and nearly negligible based on the pulse tracers from the
tropopause, while the tracers pulsed from the boundary layer show substantial inter-hemispheric transport, especially from the
NH extratropics to the SH. This is presumable because the hemispheric differences in transport mainly result from the hemi-
spheric asymmetry of the upward motion of the Hadley circulation in boreal and austral summer, of the land-sea distribution,
and of the orography, whose importance decreases with altitude. Note that the strongest cross-hemispheric transport from the
NH extratropics to the SH was diagnosed here below the tropopause at potential temperature levels 340 and 360 K. As recently
discussed in Yan et al. (2019), the Asian and North American summer monsoon tracers released at lower level (350-360 K)
and upper level (370-380 K) show similar results with upper level tracers being more confined in the NH and with lower level
tracers significantly crossing the equator.

The age spectra with respect to NH extratropical tropopause in the high latitude lower stratosphere (Hauck et al., 2020,
their Fig. 3) show a less distinct multimodal shape with much weaker seasonality compared to spectra with respect to the
NH extratropical boundary layer which are strongly affected by the seasonal variation (Fig. 5a). The age spectra for the NH
extratropical boundary origin at 60° N on the 360 K surface peak at about 2 months larger transit times compared to age spectra
related to the tropopause due to the extra vertical transport from the boundary layer to the tropopause. Both the NH extratropical
boundary air and the NH extratropical tropopause air in the NH high latitude lower stratosphere originates at the respective
surface in early summer.

AMFs from the planetary boundary layer to the troposphere were calculated by Orbe et al. (2015) using simulations from
GEOSCCM. They divided the Earth into NH, SH, and tropics regions, being 10-90° N, 10-90° S, and 10° S-10° N, respectively.
Although the domains are different to our choice here, the seasonality in the transport patterns of AMFs (their Fig. 3) shows
some similarities to our results in the UTLS (Fig. 1), e.g. the AMFs originating from the NH and SH being transported
upward mainly in summer and autumn and clear influence of Hadley cell on the tropical origin AMFs. Despite the similarities,
significant differences are found between Orbe et al. (2015) and our study. Their results show that the AMFs from the NH to
the global lower stratosphere are comparable to the corresponding contributions from the SH (their Fig. 2 and Fig. 3), while
we find that the contributions from the NH extratropics to the global lower stratosphere is about five times larger than those

from the SH extratropics. Hence, the asymmetric features of transport between NH and SH are more significant in middle-high latitude regions. The contributions from the NH and SH surface (tropics) to the atmosphere in their study are much higher (lower) than the results presented here. Crossing the subtropical tropopause transport for SH origin air happens in DJF in their study instead of MAM here. These differences are mainly attributed to the different ranges of the origin regions, the different model setup for the air mass origin in the boundary layer, and the different parameterized processes in GEOSCCM and CLaMS such as convection.

Different model setup in the boundary layer from previous studies (e.g. Orbe et al., 2016, their Fig. 5) allows us to separate the contributions from different pathways in the boundary layer and the upper level. The inter-hemispheric transport from the NH extratropics to the SH during boreal winter just happens in the boundary layer mainly caused by the convergence from the NH extratropical surface to the ITCZ region, and then the air is transported upward and southward by Hadley and BD circulations. There is almost no inter-hemispheric transport driven by the convective outflow in the high altitude during boreal winter. The cross-equatorial transport from the NH extratropics to the SH during boreal summer and autumn through UTLS is comparable to the inter-hemispheric transport through the boundary layer. Simulations from a two-box model show that the transit time of the NH origin air to the SH is shorter than vice versa associated with the different strength of the seasonal cycle and the asymmetric position of the ITCZ (Chen et al., 2017; Krol et al., 2018). However, our simulations based on three source domains show that the mean AoA from the NH extratropics surface to the SH is longer than vice versa (Fig. 6). The discrepancy is mainly caused by different definitions of the domains, which are 30-90° N and 30-90° S in our study, while Chen et al. (2017) and Krol et al. (2018) define the whole hemisphere to represent the NH and SH. Hence, cautions should be taken regarding the asymmetric transit time.

## 6   Conclusions

This paper presents AMFs and age spectra with respect to the different surface latitude bands: NH extratropics (30-90° N), SH extratropics (30-90° S), and tropics (30° S-30° N) source regions. The CLaMS model is used for carrying out simulations covering the period 1989-2017. We find that air originating at the NH extratropical surface shows about five times larger amounts in the UTLS compared to air from the SH surface. Although the tropical origin air dominates the atmosphere, the amplitude of seasonal variation is comparable for transport from the tropics and from the NH extratropics.

Both the SH extratropics and NH extratropics age spectra show more pronounced seasonality compared to the seasonality of tropical age spectra. Air in the northern high latitude regions originating from the NH extratropics is mainly transported into the UTLS from early summer to autumn, making this season particularly important for transport of anthropogenic pollutants into high latitude regions. The transit time of NH extratropical origin air to the SH extratropics is longer than vice versa, although the ASM helps to reduce this transit time.

Further analyses suggest that the cross-hemispheric transport of fresh air (AoA less than 3 months) from the NH extratropics to the SH mainly occurs in the altitude range of 320-420 K. The ASM circulation has been recognized as an important driver for cross-hemispheric transport in simulations (e.g. Orbe et al., 2016) and observations, i.e. aerosol data from wildfire plume (e.g.

Kloss et al., 2019) and volcanic plume (e.g. Wu et al., 2017). In agreement with these previous studies, we confirm the crucial role of the ASM circulation during summer in causing the cross-equatorial transport. The westerly ducts have been reported as another driver of inter-hemispheric transport during winter (e.g. Webster and Holton, 1982; Tomas and Webster, 1994). Here, we find that the westerly ducts during summer and autumn also allow inter-hemispheric transport, although they are much weaker than during winter. However, it is neither the ASM circulation nor the westerly ducts alone, but the interplay between

ASM and westerly ducts that matters for the inter-hemispheric transport from summer to autumn in the UTLS between 350 K and 370 K. In particular, it is not the region of strongest westerly ducts (Pacific) which allows the strongest transport but the Atlantic region, where the westerlies are weaker, but which is closer to the ASM, coupling with the ASM and causing strongest cross-equatorial transport.

*Data availability.* The CLaMS model outputs may be obtained from the authors upon request.

*Author contributions.* XY analyzed the data. FP carried out the model simulations. PK and FP contributed to the design of the analysis. MH contributed codes for the analysis. PK, FP, MH, AP provided helpful discussions and comments. XY wrote the paper with contributions from all co-authors.

*Competing interests.* The authors declare that they have no conflict of interest.

*Acknowledgements.* We thank the ECMWF for providing ERA-Interim meteorological reanalysis data for this study. We gratefully ac-
445 knowledge the computing time for the CLaMS simulations granted through VSR project ID JICG11 on the supercomputer JURECA at Forschungszentrum Jülich. The authors would also like to thank Darryn Waugh and the three anonymous reviewers for their very insightful comments. This research has been supported by the National Key Research and Development Program of China (grant no. 2018YFC1505703), National Natural Science Foundation of China project (grant nos. 41905040, 91837311, and 91937302), and a joint DFG-NSFC research project (DFG grant no. 392169209 and NSFC grant no. 20171352419). Felix Ploeger was funded by the Helmholtz
Association under grant no. VH-NG-1128 (Helmholtz Young Investigators Group A-SPECi).

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
