# Peer review of "Asymmetry and pathways of inter-hemispheric transport in the upper troposphere and lower stratosphere"

_Atmospheric Chemistry and Physics, 2020_

## Referee Comment (RC1) · Darryn Waugh (Referee) · 8 Dec 2020

This manuscript examines the interhemispheric transport from the surface to upper troposphere / lower stratosphere in the other hemisphere using simulations from the CLaMS model. The manuscript contains material that is of interest to ACP readers, and I think contains some new results that warrant publication. However, major revisions are required to the manuscript before it is suitable for publication. As described below, there needs to be (1) improved referencing and discussion of previous studies, (2) more precise discussion of transport in lower stratosphere versus that in upper troposphere, and (3) clearer statements on what is new (as opposed to confirming previous studies).

MAJOR COMMENTS

(1) There needs be referencing and discussion of previous studies. This applies to both the description of the method and the results from your analysis.

(a) There are many studies before Ploeger and Birner (2016) that have used boundary impulse or air mass fraction calculations (e.g., Holzer et al. 2003, Haine et al. 2008, Li et al. 2012, Orbe et al 2013, 2016). I think it is OK to refer to Ploeger and Birner (2016) for details of implementation used, but it needs to be acknowledged that others had developed similar methods and even some had used to look at similar transport problems (e.g. Orbe et al 2015, 2016). You might need to also discuss difference in implementation (see point 4 blow).

(b) There are some previous relevant transport studies that are not referenced, e.g. Holzer 2009, Orbe et al 2015. Orbe et al (2015) is particularly relevant as it addresses the same issue, and direct comparisons can be made (eg. fig 3 in Orbe et al (2015) can compared with fig 1). The issue of direct comparisons also applies to some of the papers that are already referenced, e.g. compare fig 5 of Orbe et al. 2016 with fig 7 (more on this below). It is notable that the discussion section compares with previous studies, but only those by the authors of this manuscript.

(2) There needs to be more precise discussion of transport in lower stratosphere versus that in upper troposphere. In many cases I think statements on IHT apply for transport into southern lower stratosphere but it is not clear to me that they apply for IHT into the southern upper troposphere (or more generally southern troposphere). I think you need to separate into LS or UT, or maybe be clearer on the potential temperature surfaces that a certain result applies too. For example, do statement about magnitude of air from NH compared to SH hold for both the LS and UT?

I think this separation is particularly important as the majority of the Introduction (i.e. lines 26 to 75) discusses studies of inter-hemispheric transport within the troposphere (usually NH surface to SH surface), but most of the focus of this study is on transport into lower stratosphere, and it is not clear how relevant the results are for inter hemisphere tropospheric transport. In other words, the Introduction discusses in detail previous studies of troposphere to troposphere IHT but the results from this study are not put in context of these previous studies.

(3) There needs to be clearer statements on what is new and what is confirming previous studies. The abstract contains several statements on the variation in the transport, but are these new results? Given the overlap with previous studies and limited mention or detailed comparison with these studies it is not clear which of these statements are new and which are just confirming previous studies. I think it is a bit of both, and this needs to be made clearer.

(4) One aspect that I think is new is the lack of IHT during northern winter. However, I am not sure if this not an artifact of the experiment design.

The results show virtually no transport to 10S in Dec and May (Fig 7a, h) (and according to text same for Jan-Apr). line 270-). This is very surprising, and not what is seen in other studies. I think there are many studies that show there is some IHT during NH winter. The most direct comparison is probably Orbe et al. (2016). The BIR calculations shown in fig 5 of Orbe et al. (2016) shows transport during NH winter that is similar magnitude to the summer. The summer transport in lower panel fig 5 of Orbe et al. (2016) actually looks very similar to fig 7d-e (and shows transport in monsoon and ducts), but the winter transport is very different in this manuscript. This is a clear example of a case where current results are not compared with previous studies by other authors.

Is this because a differences in the transport within your model that in previous studies or is it the method used? I think it may be the latter, as the setting boundary layer values = 0 outside the source regions means that near-surface transport south from NH source region is removed, i.e. if air is transported south near the surface before being lifted into free troposphere it will not be included in your IHT. Whether this is the

case or not, there needs to be some more discussion of the lack of winter IHT and the reason for this (and inconsistencies with previous studies).

(5) The potential interplay between the ASM and westerly ducts is I think one of the potentially new results. However, I think some care is needed in discussing this. The upper tropospheric westerly ducts are in NH winter and there are typically UT easterlies throughout the tropics in summer (as fig 9a,c shows), and I think most of previous studies on ducts and transport have focused on the winter. You are not seeing this winter transport so I don't think it is fair to say your results are in agreement in this regard (line 359). Also, the existence of summer-spring westerly ducts appears to be altitude dependent (Fig 9) and so interplay might apply in LS but not UT. Also, your statements regarding interplay between the ASM and westerly ducts could be misread to be saying the summer ASM interplay with winter westerly ducts.

MINOR COMMENTS

Figs 1-4: The changing of the scaling used for NH, SH and tropics between these figures gets confusing. I think a reader could very easily compare between columns without seeing this scaling, and once they see this in one figure they may assume similar in next figures (At least that is what I did). It might be better to have separate color bars for each column. Figures will look the same but will I think be clearer from multiple bars that scale differs.

Fig 6 What are the contours in these plots? They differ between each panel. Shouldn't the climatology be the same in each column?

Line 269 "lager"

REFERENCES

Haine, T. W. N., Zhang, H., Waugh, D. W., and Holzer, M.: On transit time distributions in unsteady circulation models, Ocean Mod., 21, 35–45, 2008

Holzer M. 2009, The path density of interhemispheric surface-to-surface transport. Part

II: Transport through the troposphere and stratosphere diagnosed from NCEP data Journal of the atmospheric sciences 66 (8), 2172-2189

Li, F., Waugh, D. W., Douglass, A. R., Newman, P. A., Pawson, S.,Stolarski, R. S., Strahan, S. E., and Nielsen, J. E.: Seasonal variations in stratospheric age spectra in GEOSCCM, J. Geophys.Res., 117, D05134, doi:10.1029/2011JD016877, 2012

Orbe, C., Holzer, M., Polvani, L.M. and Waugh, D., 2013. Air‐mass origin as a diagnostic of tropospheric transport. Journal of Geophysical Research: Atmospheres, 118(3), pp.1459-1470.

Orbe, C, DW Waugh, PA Newman 2015, Air‐mass origin in the tropical lower stratosphere: The influence of Asian boundary layer air Geophysical Research Letters 42 (10), 4240-4248

---

## Referee Comment (RC2) · Anonymous Referee #2 · 30 Dec 2020

**Summary**: The authors present a model study of zonally resolved pathways of interhemispheric transport (IHT) for air originating at the surface and arriving in the NH/SH upper troposphere and lower stratosphere. The contribution of different source regions (NH, SH, tropics) is quantified using air mass fractions for inert trace gas pulses in either region. It is concluded that the Asian summer monsoon affects IHT from the NH to the SH by interacting with westerly ducts, driving an interhemispheric asymmetry in IHT.

**Major comments**:

[Figure]

1. **The motivations/justifications for the study remain somewhat elusive**, making the novelty of the study and of the results unclear at times. The introduction lists mechanisms that have been proposed to explain IHT; what is new in the current approach? Does the current approach confirm existing literature, or does it expand on it? The introduction does not make a particularly good case for the need to answer the questions listed at L79-80. In order to make the study accessible to more readers, it may be useful to answer these basic/naive questions:

   – Are processes linked to IHT expected to be different from processes linked to transport within a given hemisphere?
   – What is the magnitude of the bias introduced to the composition of e.g. the stratosphere by the assumption that NH and SH have the same boundary conditions?

   After the comments on non zonally averaged pathways in the introduction, it was unexpected to be presented with zonally averaged pictures in the first figure (figure which is associated with commentary about non zonally averaged pathways nonetheless). Some reorganization would be helpful. Side note: I would stay away from listing the Hadley circulation itself as a mechanism for IHT (L49)– rather, I would mention the migration of the ITCZ across the equator.

2. **Some aspects of the methods and how they are applied lack discussion**. Much needed details in the methods are glossed over. Setting tracer concentrations to zero in the boundary layer outside the region of origin (L103) necessarily means that IHT pathways that go through the boundary layer will not be visible in the AMFs. This is an issue for direct comparison with previous literature, and it must be discussed. Is it possible to leave the boundary layer unperturbed, or perhaps to use parcel trajectories to address this caveat?
   What are the decay rates used for the 40 inert tracers (L100)? Do the different transit times mentioned at L104 refer to different transit time *distributions* implied

by different decay rates?

The tropics and extratropics are often separated using $30^oN/S$, but some discussion of the reasons why this threshold is used in this study is still needed. $30.0^o$ is an arbitrary number after all. Some discussion about regions where $30^oN/S$ is more or less suitable to separate tropics from extratropics would also be welcome. How much would your results vary if using e.g. $25^oN/S$? Would there be substantial benefits/caveats to defining the tropics using a dynamical, zonally resolved boundary?

3. **The interpretation of some results needs more discussion**: in the present model setup, there is virtually no NH to SH transport during Dec-May (figure 7 and associated text). Why is this? This result differs from existing literature (see e.g. Orbe et al. 2016) and needs to be further discussed, especially in light of the model setup having zero tracer concentration in the boundary layer outside the NH source region.

   Given the maps in figure 9, the coupling between the ASM and westerly ducts must be altitude dependent with transport occurring either in the upper troposphere or in the lower stratosphere. This distinction is generally absent in the study and should be included.

4. In general, I would raise the question whether the proposed approach with AMFs allows to state mechanisms the way they are, e.g. L362-363 "[. . .] coupling with the ASM and causing strongest cross-equatorial transport". **Can we be sure that it is the coupling between the westerly ducts and the ASM that causes IHT, only using maps of AMFs and meteorological composites?** Perhaps more discussion would help clear this up.

**Minor edits/suggestions** (At random and non exhaustive for the time being. More comments can be provided on a revised manuscript):

– L17: "the ozone" → "ozone".

– L21-22: "Although most [. . .] by the BD circulation" is redundant with the previous sentence.

– L22-25: "significant contributions [. . .] Wu et al., 2018)" this sentence needs a verb.

– L32-33: "the anthropogenic" → "anthropogenic", "the natural" → "natural".

– L68-69: "showed that the mean AoA [. . .] is around 1.4 years." Do you mean that the mean difference in AoA between NH and SH near-surface is 1.4 years?

– L79: "preferential/favored" pick one.

– L83: "tropics" → "the tropics". "NH extratropics" → "the NH extratropics".

– L100: "120 inert pulse trace gas species" → "120 pulses of inert trace gas species".

– L123: "total sum" → "sum".

– L132: Figure 1 shows zonally averaged results, yet the Asian summer monsoon is discussed here. This is in line with my comment about the motivations for the study: if mechanisms have been proposed to explain this zonally average picture, what is new here?

– L171: same comment as L132.

– L173: the striking feature in Figure 3 did not strike me until L173!

– L344: "patches" → "latitude bands".

– L351: "rendering" → "making" or rephrase using "granting".

**Comments on figures**:

- Generally: all captions could be shortened significantly and made clearer (see examples below). Scaling and offset factors might advantageously be avoided or made much more obvious. Many axis labels are actually repeated in white font, which takes up space that could rather be used to make the figures more legible. See for instance Figure 1 and 3 where the "Latitude [deg]" label is repeated in panels (a) through (i).

- Figure 1: please use a color scheme that does not saturate as much. To save space and make the figure clearer: put letter labels inside the panels and do not repeat the y-axis labels on each panel. Use larger coordinate ticks on all axes. I initially did not realize that the color scheme was scaled/offset for each region, and I think this may be a source of confusion for the reader. Using separate color scheme may be necessary. Otherwise, show the scaling factor ($\times 0.2$) and the offset factor (+94) in bold, much larger font. I suggest showing "$\times 1$" for the NH. The caption could be shortened by removing information made available in the text or in the figure itself: "Average zonal mean AMF (1999-2017) originated from the NH, SH, and tropics (columns) for each season (rows). The white line is the WMO tropopause. The color scheme is for the NH; scaling and offset factors are provided for the SH and tropics. AMFs for each region add up to $\sim 1$."

- Figure 2: add a colorbar and units for the streamfunction. Line up the black contours with the color shading for clarity. The title can be shortened to something like "Average residual mean mass streamfunction (1999-2017, color shading with a subset of values highlighted in black contours). The grey line is the WMO tropopause. White contours show potential temperature levels in kelvins. The black arrows illustrate the upwelling in the Hadley circulation and the shallow branch of the BDC."

- Figure 3: same comments as for Figure 1. For the caption I would suggest "Average (1999-2017) zonal mean AMF departure from the annual average. Note

the scaling factor for the SH (0.2). The absolute AMF contributions from Figure 1 are shown as black contours. Grey contours show the mean zonal winds. The black line is the WMO tropopause."

- Figure 8: highlighting the equator with a colored line would be useful.

- Figure 9: using less color shades would help read the wind speed map.

**Recommendation**
The manuscript presents an interesting approach to IHT and the mechanisms/couplings driving it. In light of my comments I suggest accepting the manuscript for publication after major revisions focused on clarifying and further discussing the methods and on improving the figures. Further improvements to the text clarity and concision can be included once major revisions are submitted.

---

## Author Comment (AC1) · 2 Mar 2021

**Referee1**

Many thanks to the reviewer for the comments which have helped to improve the clarity of the manuscript. In the following, we address all the points raised in the review (denoted by italic letters). Text changes in the manuscript are highlighted in red or blue.

*This manuscript examines the interhemispheric transport from the surface to upper troposphere / lower stratosphere in the other hemisphere using simulations from the CLaMS model. The manuscript contains material that is of interest to ACP readers, and I think contains some new results that warrant publication. However, major revisions are required to the manuscript before it is suitable for publication. As described below, there needs to be (1) improved referencing and discussion of previous studies, (2) more precise discussion of transport in lower stratosphere versus that in upper troposphere, and (3) clearer statements on what is new (as opposed to confirming previous studies).*

A. We thank the reviewer for being critical about our discussion of previous studies and statements on the results. We improve these discussions and statements in the revised manuscript as suggested by including more literature and comparisons with previous work and clarifying the statements regarding different altitude and findings.

**Major comments**

(1) *There needs be referencing and discussion of previous studies. This applies to both the description of the method and the results from your analysis.*

(a) *There are many studies before Ploeger and Birner (2016) that have used boundary impulse or air mass fraction calculations (e.g., Holzer et al. 2003, Haine et al. 2008, Li et al. 2012, Orbe et al 2013, 2016). I think it is OK to refer to Ploeger and Birner (2016) for details of implementation used, but it needs to be acknowledged that others had developed similar methods and even some had used to look at similar transport problems (e.g. Orbe et al 2015, 2016). You might need to also discuss difference in implementation (see point 4 blow).*

A. More references are added in the revised version of the manuscript. The difference between the methods is also briefly explained on P4 L107-109 as "We apply the boundary impulse (time-) evolving response (BIER) approach to calculate the age spectrum G following Ploeger and Birner (2016), which is based on the boundary impulse response (BIR) method (e.g. Holzer et al., 2003; Haine et al., 2008; Li et al., 2012; Orbe et al., 2016), but evolves with time in a transient simulation using quasi-observational winds.".

(b) *There are some previous relevant transport studies that are not referenced, e.g. Holzer 2009, Orbe et al 2015. Orbe et al (2015) is particularly relevant as it addresses the same issue, and direct comparisons can be made (eg. fig 3 in Orbe et al (2015) can compared with fig 1). The issue of direct comparisons also applies to some of the papers that are already referenced, e.g. compare fig 5 of Orbe et al. 2016 with fig 7 (more on this below). It is notable that the discussion section compares with previous studies, but only those by the authors of this manuscript.*

A. We agree that the mentioned papers are relevant for our study and extended the literature discussion as suggested. As the reviewer suggested, we compare now our Fig.1 with Fig.3 in Orbe et al (2015). We also directly compare our Fig.7 and Fig.8 with Fig.5 and Fig.7 in Orbe et al (2016). We discuss the similarities and explain the difference from previous work in the revised manuscript (P14, P17, and P18). More comparisons with the literature are also included in the "Discussion" section (P20 and P21).

(2) *There needs to be more precise discussion of transport in lower stratosphere versus that in upper troposphere. In many cases I think statements on IHT apply for transport into southern lower stratosphere but it is not clear to me that they apply for IHT into the southern upper troposphere (or more generally southern troposphere). I think you need to separate into LS or UT, or maybe be clearer on the potential temperature surfaces that a certain result applies too. For example, do statement about magnitude of air from NH compared to SH hold for both the LS and UT?*

*I think this separation is particularly important as the majority of the Introduction (i.e. lines 26 to 75) discusses studies of inter-hemispheric transport within the troposphere (usually NH surface to SH surface), but most of the focus of this study is on transport into lower stratosphere, and it is not clear how relevant the results are for inter hemisphere tropospheric transport. In other words, the Introduction discusses in detail previous studies of troposphere to troposphere IHT but the results from this study are not put in context of these previous studies.*

A. We thank the reviewer for pointing to the differences between UT and LS transport and agree that some statements don't apply for both LS and UT. We revise the conclusions which are not applicable for both UT and LS. However, we refrain from separating into UT and LS as we regard the change from UT to LS more as a gradual transition than as a distinct boundary, in particular in the tropics within the tropical tropopause layer(TTL). We clarify this point in the revised version. Most of our conclusions apply to transport in the TTL. We don't investigate the IHT in the lower troposphere due to our model setup. We mainly focus on the IHT through the TTL region. Thus, our study can be understood as a complementary approach to that in Orbe et al., 2016, and we tried to make this complementary aspect clearer in the revised manuscript.

The studies cited in the introduction section from lines 26 to 75 are mainly for emphasizing the important role of inter-hemispheric transport in regulating the distribution of atmospheric compositions, we include few studies about the impact of inter-hemisphere transport on the stratosphere. Most of previous studies are related to the inter-hemispheric transport in the troposphere. Our main goal is to extend the research to stratosphere, which is also important due to the chemical, radiative, and climate effect of atmospheric species in the stratosphere. The transport within the lower troposphere (or from NH surface to SH surface) is beyond the scope of this paper. We revised the introduction section to make the structure of the manuscript more logical.

(3) *There needs to be clearer statements on what is new and what is confirming previous studies. The abstract contains several statements on the variation in the transport, but are these new results? Given the overlap with previous studies and limited mention or detailed comparison with these studies it is not clear which of these statements are new and which are just confirming previous studies. I think it is a bit of both, and this needs to be made clearer.*

A. Detailed comparisons are included in the revised manuscript to clarify which results are new compared

to previous studies. We revise the statements mainly in the "Abstract" and "Conclusions" sections (P1 L12-14 and P22 L430-440). It is revised like "We confirm the important role of ASM and westerly ducts in the inter-hemispheric transport from the NH extratropics to the SH. Furthermore, we find that it is an interplay between the ASM and westerly ducts which triggers such cross-equator transport from boreal summer to fall in the UTLS between 350 K and 370 K." in the abstract.

(4) *One aspect that I think is new is the lack of IHT during northern winter. However, I am not sure if this not an artifact of the experiment design.*

*The results show virtually no transport to 10S in Dec and May (Fig 7a, h) (and according to text same for Jan-Apr). line 270-). This is very surprising, and not what is seen in other studies. I think there are many studies that show there is some IHT during NH winter. The most direct comparison is probably Orbe et al. (2016). The BIR calculations shown in fig 5 of Orbe et al. (2016) shows transport during NH winter that is similar magnitude to the summer. The summer transport in lower panel fig 5 of Orbe et al. (2016) actually looks very similar to fig 7d-e (and shows transport in monsoon and ducts), but the winter transport is very different in this manuscript. This is a clear example of a case where current results are not compared with previous studies by other authors.*

*Is this because a differences in the transport within your model that in previous studies or is it the method used? I think it may be the latter, as the setting boundary layer values = 0 outside the source regions means that near-surface transport south from NH source region is removed, i.e. if air is transported south near the surface before being lifted into free troposphere it will not be included in your IHT. Whether this is the case or not, there needs to be some more discussion of the lack of winter IHT and the reason for this (and inconsistencies with previous studies).*

A. Thanks for this good remark! Yes, we think that the lack of IHT during northern winter is likely caused by the model setup in our simulations with the species set to zero in the boundary layer outside of the origin region, which eliminates the cross-equatorial transport from NH extratropical surface to the intertropical convergence zone (ITCZ) and then ascent over the tropics and SH subtropics. The suppressed convection in NH (sub)tropics in boreal winter can not lift the NH extratropical air to high altitude likely causing the lack of inter-hemispheric transport during winter. In our study, transport out of the NH extratropics and into the tropics in the lowest model layer is forbidden, and subsequent ascent is referred to as transport from the tropical surface. This is clearly a difference to the study by Orbe et al. (2016), but we don't think that one of the approaches can be regarded as better - they are just complementary, focusing on different aspects of transport. We include the comparison with previous studies and explanations of the differences in the revised manuscript.

(5) *The potential interplay between the ASM and westerly ducts is I think one of the potentially new results. However, I think some care is needed in discussing this. The upper tropospheric westerly ducts are in NH winter and there are typically UT easterlies throughout the tropics in summer (as fig 9a,c shows), and I think most of previous studies on ducts and transport have focused on the winter. You are not seeing this winter transport so I don't think it is fair to say your results are in agreement in this regard (line 359). Also, the existence of summer-spring westerly ducts appears to be altitude dependent (Fig 9) and so interplay might apply in LS but not UT. Also, your statements regarding interplay between the ASM and westerly ducts could be misread to be saying the summer ASM interplay with winter westerly ducts.*

A. We agree that the coupling between the ASM and westerly ducts depends on the altitude and the season. The interaction between ASM and westerly ducts mainly drives the inter-hemispheric transport during autumn in the UTLS due to the coincidence of the westerly ducts and a significant amount of NH air at UTLS levels, which was transported upwards by the ASM during the previous months. This coupling effect plays an important role in the inter-hemispheric transport from summer to autumn at the altitude level between 350 K and 370 K. Our statements before, indeed might have been misleading and we revised the statements about the results and conclusions accordingly (P18, P19, and P22).

**Minor comments**

(1) *Figs 1-4: The changing of the scaling used for NH, SH and tropics between these figures gets confusing. I think a reader could very easily compare between columns without seeing this scaling, and once they see this in one figure they may assume similar in next figures (At least that is what I did). It might be better to have separate color bars for each column. Figures will look the same but will I think be clearer from multiple bars that scale differs.*

A. The colorbar for each source region is now included in the figures in the revised version.

(2) *Fig 6 What are the contours in these plots? They differ between each panel. Shouldn't the climatology be the same in each column?*

A. The contours represent the mean age of air from each source region (NH extratropics, SH extratropics, and tropics) to the atmosphere, hence they are different in each column.

(3) *Line 269 "lager"*

A. It's corrected.

---

## Author Comment (AC2) · 2 Mar 2021

**Referee2**

Many thanks to the reviewer for the comments which have have helped to improve the clarity of the manuscript. In the following, we address all the points raised in the review (denoted by italic letters). Text changes in the manuscript are highlighted in red or blue.

**Summary:** *The authors present a model study of zonally resolved pathways of interhemispheric transport (IHT) for air originating at the surface and arriving in the NH/SH upper troposphere and lower stratosphere. The contribution of different source regions (NH, SH, tropics) is quantified using air mass fractions for inert trace gas pulses in either region. It is concluded that the Asian summer monsoon affects IHT from the NH to the SH by interacting with westerly ducts, driving an interhemispheric asymmetry in IHT.*

**Major comments**

1 *The motivations/justifications for the study remain somewhat elusive, making the novelty of the study and of the results unclear at times. The introduction lists mechanisms that have been proposed to explain IHT; what is new in the current approach? Does the current approach confirm existing literature, or does it expand on it? The introduction does not make a particularly good case for the need to answer the questions listed at L79-80. In order to make the study accessible to more readers, it may be useful to answer these basic/naive questions:*

*– Are processes linked to IHT expected to be different from processes linked to transport within a given hemisphere?*

*– What is the magnitude of the bias introduced to the composition of e.g. the stratosphere by the assumption that NH and SH have the same boundary conditions?*

*After the comments on non zonally averaged pathways in the introduction, it was unexpected to be presented with zonally averaged pictures in the first figure (figure which is associated with commentary about non zonally averaged pathways nonetheless). Some reorganization would be helpful. Side note: I would stay away from listing the Hadley circulation itself as a mechanism for IHT (L49)– rather, I would mention the migration of the ITCZ across the equator.*

A. We agree that the link between the introduction and questions addressed in the study was not so clear. We revised the introduction entirely, including the Reviewer's remarks, and the questions addressed in our study.

We present the results from both zonally resolved and zonally averaged calculations to address different questions. To get the global view of the contributions from the three source regions, we present the zonally averaged results. Later we present the zonally resolved results in Fig. 7 and Fig. 8 to explore the pathways of inter-hemispheric transport. We reorganized the text and tried to clarify the purpose of using both zonal mean and zonally resolved diagnostics in the revised manuscript.

Also the formulation regarding the mechanism related to Hadley cell is revised as suggested.

2 *Some aspects of the methods and how they are applied lack discussion. Much needed details in the methods are glossed over. Setting tracer concentrations to zero in the boundary layer outside the region of origin (L103) necessarily means that IHT pathways that go through the boundary layer will not be*

*visible in the AMFs. This is an issue for direct comparison with previous literature, and it must be discussed. Is it possible to leave the boundary layer unperturbed, or perhaps to use parcel trajectories to address this caveat?*

*What are the decay rates used for the 40 inert tracers (L100)? Do the different transit times mentioned at L104 refer to different transit time distributions implied by different decay rates?*

*The tropics and extratropics are often separated using 30° N/S, but some discussion of the reasons why this threshold is used in this study is still needed. 30.0° is an arbitrary number after all. Some discussion about regions where 30° N/S is more or less suitable to separate tropics from extratropics would also be welcome. How much would your results vary if using e.g. 25° N/S? Would there be substantial benefits/caveats to defining the tropics using a dynamical, zonally resolved boundary?*

A. Yes, we agree that our model setup in the boundary layer is different from previous studies. In our study, the pulse tracer mixing ratios are set to one in the boundary layer of the source region, and are set to zero in the boundary layer outside of the initialization region in every time step. This boundary condition eliminates the cross-equatorial transport from the NH extratropical surface to the intertropical convergence zone (ITCZ) and then ascent over the tropics and SH subtropics as discussed in Orbe et al., 2016. On the other hand, our boundary condition allows us to separate the contribution of the direct ascent in the NH and of the related IHT in the upper levels (mainly through the TTL). We show that the related cross-equatorial transport from the NH extratropics to the SH during boreal summer and fall driven by the convective outflow, Asian monsoon, and westerly ducts contributes more than half of the total abundance of NH extratropical air in the UTLS in the SH. The relative importance of the inter-hemispheric transport through the ASM pathway compared to those over Atlantic and Pacific seems to be stronger in our study than in Orbe et al. (2016, c.f. their Fig. 5f-j). Although the model setup is different, we find some similarities between previous studies and our results. We include detailed comparisons to previous literature in the new manuscript (P14, P17, P18, P20, and P21).

The tracers used in our study are inert pulse tracers without decay. Different transit times mean that the boundary tracers are pulsed in different time.

The threshold of 30.0° is a common choice to separate tropics from extratropics (e.g. Fueglistaler et al., 2011) as it divides the Earth's surface into equal areas of both regions, and further coincides approximately with the horizontal transport barriers of the subtropical jet cores. We include the reason in the revised manuscript. The results from different domains around 30° N/S (e.g. 25° N/S) require sensitivity studies, which are computationally intensive for our climatological study. We compare our results with Orbe 2015 et al, who separate the domains to NH (10-90° N), SH (10-90° S), and tropics (10° S-10° N), and find that the contributions from the NH to the global stratosphere are comparable to those from the SH. Hence, the asymmetric features are more significant in high latitude. Defining the tropics with dynamical and zonally resolved boundary is beneficial to study the transport from the tropics to the global atmosphere, which is also computationally intensive and beyond the scope of this paper. We mainly focus on the inter-hemispheric transport between NH extratropics and SH extratropics.

3 *The interpretation of some results needs more discussion: in the present model setup, there is virtually no NH to SH transport during Dec-May (figure 7 and associated text). Why is this? This result differs from existing literature (see e.g. Orbe et al. 2016) and needs to be further discussed, especially in light of the model setup having zero tracer concentration in the boundary layer outside the NH source region.*

*Given the maps in figure 9, the coupling between the ASM and westerly ducts must be altitude dependent with transport occurring either in the upper troposphere or in the lower stratosphere. This distinction is generally absent in the study and should be included.*

A. The AMF from December to May is nearly zero in our study, which is different from the previous results (e.g. Holzer, 2009 and Orbe et al, 2016). This is likely caused by the model setup in our simulations with the species set to zero in the boundary layer outside of the origin region, which eliminates the cross-equatorial transport from NH extratropical surface to the ITCZ region and then ascent over the tropics and SH subtropics. The suppressed convection in NH (sub)tropics in boreal winter can not lift the NH extra-tropical air to high altitude likely causing the lack of inter-hemispheric transport during winter. Detailed comparisons are included in the revised manuscript (P14, P17, and P18).

We agree that the coupling effect between ASM and westerly ducts is time and altitude dependent. The interaction between ASM and westerly ducts mainly drives the inter-hemispheric transport during autumn in the UTLS due to the coincidence of the westerly ducts and a significant amount of NH air at UTLS levels, which was transported upwards by the ASM during the previous months. This coupling effect plays an important role in the inter-hemispheric transport from summer to autumn at the altitude level between 350 K and 370 K. The coupling effect between ASM and westerly ducts during summer is different from the interaction during autumn. The interaction between ASM and westerly ducts happens at the same time during summer and transports the NH extratropical air to the SH. The coupling effect during autumn refers to the transport driven by the ASM during summer and the inter-hemispheric transport through westerly ducts during autumn. We separate the statements regarding different altitudes and season and include this point in the revised manuscript (P18 and P19).

4 *In general, I would raise the question whether the proposed approach with AMFs allows to state mechanisms the way they are, e.g. L362-363 "[. . .] coupling with the ASM and causing strongest cross-equatorial transport". Can we be sure that it is the coupling between the westerly ducts and the ASM that causes IHT, only using maps of AMFs and meteorological composites? Perhaps more discussion would help clear this up.*

A. Indeed, it is difficult to prove the existence of the coupling in a strict way, however we think the presented results provide strong evidence for its existence. We state the cross-equatorial transport driven by the coupling of ASM and westerly ducts based on the evolution of AMFs. We also include a new figure (Figure RL 1) in the revised manuscript to depict the coupling effect. Westerly ducts in the UTLS (350-380 K) are strongest in January and weakest in September (please see Figure RL 1). However, we find that the largest inter-hemispheric transport happens in September (please see Fig. 7 in the manuscript) with neither the strongest ASM nor the strongest westerly ducts. The largest amount of NH extratropical air (highest AMFs) are transported to the UTLS over southern ASM region, Pacific, and Atlantic in September by the ASM circulation and eddy shedding before crossing the equator (Figure RL 1). And they are larger than the AMFs in July (strongest ASM) and in January (strongest westerly ducts), which again suggests that neither the westerly ducts nor the ASM alone determines cross-equatorial transport.

Figure RL 2, Figure RL 3, and Figure RL 4 respectively show the monthly evolution of young (< 3 months) AMFs on the 340 K, 370 K, and 420 K isentropic surface from May to December. Almost no air can be driven by the westerly ducts directly from the NH extratropical surface into the SH bypassing the ASM. All

[Figure]

Figure RL 1: Longitude-potential temperature cross-section along the equator of monthly mean zonal wind over-plotted with the monthly zonal mean of NH extratropical young (< 3 months) AMF (contours) over the domain of [6° N, 20° N] during 1999-2017.

the NH extratropical air over the westerly ducts in the east Pacific and east Atlantic is originally from the ASM region few months before. But this coupling effect is time and altitude dependent like we discussed above. The interaction between ASM and westerly ducts mainly drives the inter-hemispheric transport during autumn in the lower (e.g. 340 K) and upper edge (e.g. 420 K) of the ASM anticyclone. The coupling effect works from summer to autumn in the altitude with strong ASM anticyclone (350 K-370 K).

**Minor edits/suggestions** *(At random and non exhaustive for the time being. More comments can be provided on a revised manuscript):*

– *L17: "the ozone" → "ozone".*

A. It's revised.

– *L21-22: "Although most [. . .] by the BD circulation" is redundant with the previous sentence.*

A. It's revised.

– *L22-25: "significant contributions [. . .] Wu et al., 2018)" this sentence needs a verb.*

A. The verb is added.

[Figure]

Figure RL 2: A snapshot of the monthly mean horizontal distribution of the NH extratropical origin young (< 3 months) air on the 340 K isentropic surface from May to December. Streamlines show horizontal winds.

– **L32-33: "the anthropogenic" → "anthropogenic", "the natural" → "natural".**

A. It's removed.

– **L68-69: "showed that the mean AoA [. . .] is around 1.4 years." Do you mean that the mean difference in AoA between NH and SH near-surface is 1.4 years?**

A. We mean that the mean AoA from the NH midlatitude surface to the SH midlatitudes surface is around 1.4 years, not the mean difference between NH and SH. Although the mean difference in AoA between NH and SH near-surface is around 1.4 years in Waugh et al., 2013 based on the calculation of SF6 because the mean AoA in the NH is nearly zero.

– **L79: "preferential/favored" pick one.**

A. "preferential" is removed.

[Figure]

Figure RL 3: Same as Figure RL 2 but on the 370 K isentropic surface.

– **L83: "tropics" → "the tropics". "NH extratropics" → "the NH extratropics".**

A. "the" is added.

– **L100: "120 inert pulse trace gas species" → "120 pulses of inert trace gas species".**

A. It's revised.

– **L123: "total sum" → "sum".**

A. "total" is removed.

– **L132: Figure 1 shows zonally averaged results, yet the Asian summer monsoon is discussed here. This is in line with my comment about the motivations for the study: if mechanisms have been proposed to explain this zonally average picture, what is new here?**

[Figure]

Figure RL 4: Same as Figure RL 2 but on the 420 K isentropic surface.

A. The previous studies did not show the contributions of air from both the NH extratropics and the SH extratropics (30-90° N/S). The asymmetric features of inter-hemispheric transport between two hemispheres are more significant compared to previous studies, which highlight the difference between the NH extratropics and SH extratropics. The absolute contributions from the extratropics and the asymmetric features are new compared to previous study. Asian summer monsoon is referred to explain the asymmetric features. We confirm the impact of Asian summer monsoon on the inter-hemispheric transport and introduce our findings step by step. In the end, we propose the effect of interaction between ASM and westerly ducts.

– *L171: same comment as L132.*

A. Previous studies did not quantify and explain the impact of ASM on the transport of NH extratropical boundary air to the NH extratropical lower stratosphere. There was no study comparing the inter-hemispheric transport from the NH extratropics to the SH with the inter-hemispheric transport from the SH extratropics to the NH. The contributions and patterns of transport are also different from previous studies

despite the similarities.

– *L173: the striking feature in Figure 3 did not strike me until L173!*

A. We think that the seasonality of NH extratropical air in the global stratosphere is comparable to the seasonality of corresponding tropical air, and they are about five times of the seasonality from the SH extratropics. The positive anomaly of NH extratropical air in the NH extratropical and tropical lower stratosphere is significant. This positive anomaly also occurs in the SH subtropical troposphere. These results shown in Fig. 3 are quite new and striking. Previous studies show the contributions from the SH and NH are comparable. Our study shows more significant asymmetric features in the inter-hemispheric transport.

– *L344: "patches" → "latitude bands".*

A. It's revised.

– *L351: "rendering" → "making" or rephrase using "granting".*

A. It's revised.

**Comments on figures:**

- *Generally: all captions could be shortened significantly and made clearer (see examples below). Scaling and offset factors might advantageously be avoided or made much more obvious. Many axis labels are actually repeated in white font, which takes up space that could rather be used to make the figures more legible. See for instance Figure 1 and 3 where the "Latitude [deg]" label is repeated in panels (a) through (i).*

  A. The captions are shortened as suggested. The "Latitude [deg]" label is not repeated in panels (a) through (i) in Figure 1 and 3. Do you mean "Pot. temperature [K]"? Most figures are edited.

- *Figure 1: please use a color scheme that does not saturate as much. To save space and make the figure clearer: put letter labels inside the panels and do not repeat the y-axis labels on each panel. Use larger coordinate ticks on all axes. I initially did not realize that the color scheme was scaled/offset for each region, and I think this may be a source of confusion for the reader. Using separate color scheme may be necessary. Otherwise, show the scaling factor (×0.2) and the offset factor (+94) in bold, much larger font. I suggest showing "×1" for the NH. The caption could be shortened by removing information made available in the text or in the figure itself: "Average zonal mean AMF (1999-2017) originated from the NH, SH, and tropics (columns) for each season (rows). The white line is the WMO tropopause. The color scheme is for the NH; scaling and offset factors are provided for the SH and tropics. AMFs for each region add up to ∼1."*

  A. We tried many color schemes before, this one shows best visibility of transport and contributions from the source regions. We edit the figure and shorten the caption in the new manuscript.

- *Figure 2: add a colorbar and units for the streamfunction. Line up the black contours with the color shading for clarity. The title can be shortened to something like "Average residual mean mass stream-function (1999-2017, color shading with a subset of values highlighted in black contours). The grey line*

*is the WMO tropopause. White contours show potential temperature levels in kelvins. The black arrows illustrate the upwelling in the Hadley circulation and the shallow branch of the BDC."*

A. The figure and the caption is revised as suggested.

- *Figure 3: same comments as for Figure 1. For the caption I would suggest "Average (1999-2017) zonal mean AMF departure from the annual average. Note the scaling factor for the SH (0.2). The absolute AMF contributions from Figure 1 are shown as black contours. Grey contours show the mean zonal winds. The black line is the WMO tropopause."*

  A. The figure and the caption is revised as suggested.

- *Figure 8: highlighting the equator with a colored line would be useful.*

  A. The figure and is revised as suggested.

- *Figure 9: using less color shades would help read the wind speed map.*

  A. The figure and is revised as suggested.